# Agricultural Pest Management: The Role of Microorganisms in Biopesticides and Soil Bioremediation

**DOI:** 10.3390/plants13192762

**Published:** 2024-10-01

**Authors:** Alane Beatriz Vermelho, Jean Vinícius Moreira, Ingrid Teixeira Akamine, Veronica S. Cardoso, Felipe R. P. Mansoldo

**Affiliations:** 1Bioinovar Laboratory, General Microbiology Department, Institute of Microbiology Paulo de Goes, Federal University of Rio de Janeiro, Rio de Janeiro 21941-902, RJ, Brazil; jeanvmoreira@micro.ufrj.br (J.V.M.); ingrid.akamine@micro.ufrj.br (I.T.A.); verocardoso@micro.ufrj.br (V.S.C.); mansoldo@micro.ufrj.br (F.R.P.M.); 2Center of Excellence in Fertilizers and Plant Nutrition (Cefenp), SEDEICS, Rio de Janeiro 21941-850, RJ, Brazil

**Keywords:** biopesticides, environment, microbial origin, pest crops, biopesticide application, bioremediation

## Abstract

Pesticide use in crops is a severe problem in some countries. Each country has its legislation for use, but they differ in the degree of tolerance for these broadly toxic products. Several synthetic pesticides can cause air, soil, and water pollution, contaminating the human food chain and other living beings. In addition, some of them can accumulate in the environment for an indeterminate amount of time. The agriculture sector must guarantee healthy food with sustainable production using environmentally friendly methods. In this context, biological biopesticides from microbes and plants are a growing green solution for this segment. Several pests attack crops worldwide, including weeds, insects, nematodes, and microorganisms such as fungi, bacteria, and viruses, causing diseases and economic losses. The use of bioproducts from microorganisms, such as microbial biopesticides (MBPs) or microorganisms alone, is a practice and is growing due to the intense research in the world. Mainly, bacteria, fungi, and baculoviruses have been used as sources of biomolecules and secondary metabolites for biopesticide use. Different methods, such as direct soil application, spraying techniques with microorganisms, endotherapy, and seed treatment, are used. Adjuvants like surfactants, protective agents, and carriers improve the system in different formulations. In addition, microorganisms are a tool for the bioremediation of pesticides in the environment. This review summarizes these topics, focusing on the biopesticides of microbial origin.

## 1. Introduction

It is essential to improve environmental quality to drive economic development and enhance societal well-being and sustainable development [1]. The pollution of the environment, including air, water, and soil, has become an important and pressing issue in recent years. Greenhouse emissions, plastic pollution, toxic industrial effluents/residues, and pesticide contamination, among others, are compromising the planet and human health. In this context, the industrial sector has been actively trying to use cleaner production technologies and ecologically friendly practices. Recognizing possible pollution sources and applying preventive actions to reduce environmental impact is not just crucial but a matter of global importance to addressing global pollution [2].

Ensuring safe food is a fundamental human right and a pressing global defy, regardless of individuals’ economic and social circumstances. The agricultural sector is facing a challenge in the coming years as it strives to ensure that food production can meet the needs of the rapidly growing global population, which is projected to reach 9.7 and 10.9 billion in 2050 and 2100, respectively, according to the Food and Agriculture Organization [3,4,5].

The topic of pesticides in agriculture is of utmost importance. Despite their potential harm, pesticides play a crucial role in modern agriculture, safeguarding crops from pests, diseases, and weeds, thereby ensuring higher yields and better-quality production. The absence of pesticides would profoundly impact global food production, increasing food insecurity and higher consumer prices. Legislations in different countries regulate the use of pesticides to ensure consumer safety. These rules vary across countries, and it is unclear what the situation is for these products in national markets despite the growing awareness of pesticide risks. Government incentives drive the renewal and enhancement of regulations in major agricultural production countries [6].

However, several issues arise due to the toxicity, high persistence, and pervasiveness of these substances in the environment. Their byproducts may also linger and interact with the environment and living organisms in various ways, depending on their nature, chemical structure, dosage, and targets. This interaction can significantly alter all life forms [7].

This review encompasses aspects of the common pests in crops, focusing on microbial pesticides, including the major biopesticide-producing microorganisms and methodologies used.

## 2. Major Pest Crop

### 2.1. Weeds

The most crucial limitation of biotic factors to agricultural production is weeds. In developing and developed countries, weeds cause yield loss, which is a direct result of weeds competing for resources such as light, soil nutrients, space, carbon dioxide, and water [8,9].

Also, factors like weed emergence timing, weed density, weed types, and crops can cause yield losses. If uncontrolled, weeds can cause complete (100%) yield loss. Additionally, they serve as hosts for insects and pathogens that target crop plants [8].

It has been demonstrated that there is an increased number of weeds and herbicide-resistant species in weeds [10]. Currently, 531 cases of herbicide-resistant weeds are reported globally, and weeds are responsible for resistance in 168 herbicides [11,12].

Some are poisonous and may contain toxins from several major chemical groups, including alkaloids, glycosides, saponins, resinoids, oxalates, and nitrates [13].

Annual losses are estimated to exceed USD 26 billion in the USA, AUD 3.3 billion in Australia, and USD 11 billion in India, exclusively across 10 major crops [14]. A study in the Indian Western Himalayas revealed a staggering diversity of 59 weed species. The Asteraceae family emerged as the most diverse, with 15% of the species, followed by Poaceae (14%) and Brassicaceae (12%). The dominant life form among these weeds was Therophytes, followed by Hemicryptophytes. This diversity underscores the complexity of the weed problem, making it a challenge for agriculture [15].

### 2.2. Ticks (Miticides/Acaricides)

Mites and ticks are part of the Acari subclass within the Arachnida, which includes over 54,000 described species. Phytophagous mites are plant-feeding mites that attack nearly all cultivated crops, including ornamental and wild plants. This can significantly reduce crop yields and make them vectors for several plant diseases [16].

The citrus red mite (*Panonychus citri*) and the European red mite (*Panonychus ulmi*) are some examples of economically important pests in citrus and apple production [17].

Mites of the Tetranychidae family feed on a wide variety of vascular plants throughout the world, such as *Tetranychus urticae* and *Tetranychus cinnabarinus*, are spider pests that can thrive on more than 1000 plant species from more than 140 different plant families, including 150 different crops including tomato, citrus, *Cymbidium*, *Dianthus*, *Gladioli*, and *Tagetes* [18].

### 2.3. Plant Microbial Diseases

#### 2.3.1. Fungi

Fungi are probably the most important microbial plant pathogens. The major fungal plant pathogens are *Fusarium* spp., *Pythium* spp., *Colletotrichum* spp., *Alternaria* spp., *Plasmodiophora* spp., *Taphrina* spp., and *Puccinia* spp. [19,20,21]. Seed-borne and seed-transmitted fungal diseases have emerged as a significant threat to the vegetable seed production system. The genera Aschochyta, Didymella, Phoma, Phytophthora, Rhizoctonia, Sclerotinia, and Sclerotium have increased on vegetable crops to a great extent [22]. Some important fungal diseases are described in Table 1.

#### 2.3.2. Phytopathogenic Bacteria

Plant diseases caused by phytopathogenic bacteria constantly threaten crop production, leading to significant annual losses on a global scale [34]. They lead to symptoms such as spots, blights, cankers, tissue rots, and/or hormone imbalances that lead to plant overgrowth, stunting, root branching, and leaf epinasty, among others [35]. The genera of plant pathogenic bacteria known as problematic for crops and widely distributed are the Gram-negative Xanthomonas, Agrobacterium, Pseudomonas, Xylella, and Erwinia. Regarding Gram-positive bacteria, the genera Arthrobacter, Clavibacter, Curtobacterium, and Rhodococcus are commonly found [36]. *Ralstonia solanacearum* is a soil-borne bacteria and is the world’s second most harmful bacterial phytopathogen. It is the agent of bacterial wilt (BW) disease, affecting 310 species and 42 families of plants. This pathogen is potato crops’ second most prevalent disease [37,38]. *Dickeya solani* and Pectobacterium species are two important potato pathogens. Both are necrotrophic and cause similar symptoms, resulting in blackleg disease on stems and soft-rot disease of tubers [39].

It is estimated that bacterial infections causing plant diseases contribute to 10–16% of the total global losses in plant production. Economically, this represents over one billion US dollars annually [40].

#### 2.3.3. Virus

Plant viruses have assumed a significant economic role in crops through their devastating impact on plant health. While diseases caused by plant viruses may not be curable, applying effective management techniques can significantly aid in preventing and controlling their spread, particularly in the early stages of infection. It is important to note that controlling viral diseases is not a simple task. As with other plant diseases, it requires substantial investments in control methods such as pesticides and management strategies [41,42].

Most plant viruses that cause significant agricultural losses are transmitted by sap-sucking insect vectors, including aphids, leafhoppers, planthoppers, thrips, and whiteflies. These insect vectors are responsible for transmitting over 70% of plant viruses. The acquisition and transmission of viruses by their vectors necessitate a close interaction between viral proteins and vector-associated proteins, commonly known as receptors, and several receptors have been recognized [43]. Insect viruses are distributed into two major groups: The first group consists of the insect-specific viruses (ISVs), which exclusively replicate within insect hosts. This group includes Baculoviridae, Parvoviridae, Flaviviridae, Ascoviridae, and Togaviridae families. The second group comprises viruses that replicate in insects but can infect vertebrates or plants. This group includes members of the Togaviridae, Flaviviridae, Reoviridae, and Phenuiviridae families, as well as certain plant viruses within the Tospoviridae, Reoviridae, and Rhabdoviridae families [42].

#### 2.3.4. Microalgae

A few algae are a problem in agriculture. Usually, the microalga *Cephaleuros* spp. causes algal leaf spot disease in tree crops, such as cocoa, coffee, tea, and rubber, in the humid tropics. Orange- or brownish-colored spots with a velvet appearance are seen on the leaf surface. Leaf spots are relatively small in size, and enlargement of the spots is also not common or very slow [44]. Cephaleuros is prevalent in India, causing tea plants a disease known as the red rust of tea, a severe problem in plantations [45].

### 2.4. Insects

#### 2.4.1. Termites

Termites are insects of the order Isoptera, with about 3000 species, and around 70% of termite species belong to a single family, Termitidae [46]. They are widespread worldwide, playing a crucial role as wood decomposers, nutrient recyclers, and global carbon cycles [47]. The most critical termite genera in Africa and Ethiopia are Macrotermes, Odnotermes, Pseudocanthotermes, Ancistrotermes, and Microterme termites [47]. Many species of termites have been reported to attack maize, but *Microtermes* spp., *Macrotermes* spp., *Odontotermes* spp., and *Allodontermes* spp. were found to cause economic losses to the crop. They cause substantial losses in other important crops, such as cereals (rice, wheat, barley, millet, and sorghum), pulses, oilseeds, vegetables, fruits (guava, citrus, banana, mango, papaya, grapes, etc.), sugarcane, and cotton [48,49]. Global damage caused by termites is estimated to exceed USD 40 billion annually, with the annual economic loss in the USA alone estimated at USD 11 billion, USD 1 billion in China, USD 35.12 million in India, and USD 1 billion in Indonesia [50,51].

#### 2.4.2. Other Insects

Despite the limited availability of statistical data, arthropods may be responsible for destroying an estimated 18–20% of annual worldwide crop production, valued at more than USD 470 billion [52]. The synthetic pesticide neonicotinoid controls several of them. Some examples are the western corn rootworm (WCR), *Diabrotica virgifera* spp. virgifera LeConte (Coleoptera: Chrysomelidae), and a mixture of wireworm species such as *Agriotes* ssp. and Coleoptera: Elasteridae, common pests in maize. Another significant cereal problem is *Rhopalosiphum padi* (Linnaeus), the bird cherry-oat aphid. It transmits pathogens such as barley yellow dwarf virus and produces a high amount of honeydew, interfering with photosynthesis [53]. Another pest is the brown planthopper (BPH, *Nilaparvata lugens*) in rice, which can cause up to 70% yield losses through direct feeding and virus transmission [54]. In cotton, the cotton aphid (*Aphis gossypii*) [55], and silver-leaf whitefly (SLW, *Bemisia tabaci*) [56] damage the crop through direct feeding, virus vectoring, and the extensive secretion of honeydew [57]. It is essential to highlight that excessive use of chemical pesticides has led to the emergence of resistance, resurgence, and residues [58].

### 2.5. Nematodes

Plant-parasitic nematodes are significant soil-borne crop pests, and their control relies on highly toxic pesticides that impact the entire soil community [59]. There are more than 4100 phytoparasitic nematodes. They are classified into three groups: The first comprises endoparasites, which penetrate inside the plant, reducing its growth and productivity. The major groups are root-knot nematodes (*Meloidogyne* spp.), cyst nematodes (*Heterodera* spp., *Globodera* spp.), root lesion nematodes (*Pratylenchus* spp., *Hirschmanniella* spp., *Radopholus* spp.), stem nematodes (*Ditylenchus* spp.), and the pine wood nematode Bursaphlenchus xylophilus. In particular, root-knot nematodes (*Meloidogyne* spp.) affect crops such as watermelon, cucumber, eggplant, and tomato, reducing their yield. The second group develops outside the plant. The process involves puncturing the epidermis or the superficial layers of the host plant root cortex with the stylet. Following feeding, they migrate to another area of the root or another plant. In addition, some of them, such as *Xiphinema* spp., *Trichodorus* spp., and *Paratrichodorus* spp., can transmit viruses. Species like *Sphaeronema* spp., *Hoplolaimus* spp., and *Helycotylenchus* spp. belong to the last classification, semi-endoparasites, which can penetrate partially into the plant [59]. Plant nematodes are found in a great diversity of plants and crops worldwide, causing significant economic losses. In India, plant-parasitic nematodes cause 21.3% crop losses of USD 1.58 billion annually [60]. It is estimated that 14% of global crop yield is lost annually, amounting to USD 125 billion, as the American Phytopathological Society (APS) reported [61].

### 2.6. Rodents

There are over 2200 species of rodents known around the world [62]. Mammals, such as rabbits and ungulates, can cause crop damage. However, nearly all crops grown globally, including cereal grains, vegetables, cotton, alfalfa, sugar cane, potatoes, tree fruits, and many others, are vulnerable to rodent damage. They are commonly found in fields, orchards, farms, livestock stables, grain storage facilities, food industry sites, and various other locations within agricultural and human-made environments. Rodents can also damage crops in storage, posing a risk of contaminating food supplies with feces and urine [62,63,64].

### 2.7. Mollusks

Terrestrial mollusks (slugs and snails) are pest crops that impact some vegetables, fruit trees, tubers, ornamentals, legumes, bark, leaves, and fruits. Some examples of crops attacked by land mollusks are potatoes, cereal, lettuce, cabbage, carrots, maize, clover, and other horticultural crops [65]. They can consume over 30% of their own body weight in 24 h. Species like the giant African snail, *Achatina fulica* Bowdich, consume a wide range of organic material, including carcasses and feces from various organisms, even humans [66,67].

Holes and slimy damaged parts of the crops and vegetables characterize mollusks’ damage. Snails and slugs both have similar biology [68].

Some important species of mollusks are *Achatina fulicva*, *Helix aspersa*, *Theba pisana*, *Arianta arbustorum*, and *Argiolimax metriculatus.* Genera such as Deroceras, Tandonia, Milax, Arion, Limax, Lehmanniopoirieri, Limacusflavus, Milaxgagates, and Laevicaulisalte are frequently found [69].

## 3. Pesticides and Biopesticides

Pesticide is used to describe a range of chemicals used as pest control in agriculture. The lack of selective toxicity, biodegradability, and accumulation in the environment of these synthetic pesticides has become a significant problem for their use. The most common, such as dichlorodiphenyltrichloroethane (DDT) and toxaphene, can have long-lasting effects on the soil and may leach into groundwater and aquatic ecosystems [70,71]. The half-time of the pesticides dichlorodiphenyldichloroethylene (DDE) and dichlorodiphenyltrichloroethane (DDD) is 15 years [72].

Some pesticides, such as carbofuran, carbendazim, dichlorvos, anthraquinone, dinocap, paraquat, methomyl, aldicarb, and diuron, have been banned in many countries due to their toxicity to humans [73]. Pesticides can lead to the bioaccumulation of contaminants in food chains, which are widely spread in terrestrial and aquatic environments. When applied excessively and washed away into water bodies during rainfall, they can cause the death of fish and other aquatic life. They can also harm soil texture, microbes, animals, and plants [74].

Besides this, approximately 150,000 people die each year from pesticide poisoning. The majority of these deaths are due to self-poisoning through ingestion rather than from occupational or accidental exposures, which are typically through skin contact or inhalation. Those who consume contaminated food may indirectly ingest these chemicals, leading to biomagnification in their body systems. Serious health problems could result from this intoxication, such as cancer, kidney disease, diabetes, liver dysfunction, eczema, neurological damage, and cardiovascular disease [75]. These facts highlight the critical need for sustainable alternatives to conventional pesticides. Significant research for developing green or less toxic pesticides has increased worldwide to mitigate these risks.

In this context, biopesticides arise and comprise many substances from natural sources, including botanical, microbial, and biochemicals derived from microorganisms. Another process is incorporating DNA into agricultural products that confer protection against pest damage. Biopesticides are considered the best alternative to synthetic pesticides, as they are highly effective, target-specific, more environmentally friendly, sustainable, and reduce environmental risks. Microbial pesticides represent an advancement in pest control compared to conventional agrochemical pesticides, being efficient and biodegradable [74,76,77].

Currently, biopesticide use is growing and comprises about 10% of the global pesticide market due to restrictions on chemical pesticides, pesticide resistance, and residue management [78]. The shelf-life of biopesticides is short, and they have no residual effects on the environment. They are specific in their target, preventing bioaccumulation [79].

Pesticides and biopesticides can be classified according to (*i*) their source as chemical or biological. They are also classified depending on (*ii*) their targets, or the functions they perform. In this context, they range from those employed in the control of weeds (herbicides), algae (algicides), fungi (fungicide), mites or ticks (miticides/acaricides), bacteria (bactericides), rodents (rodenticide), termites (termiticides), insects (insecticides), mollusks (molluscicides), and nematodes (nematicides). Figure 1 summarizes the pests that are relevant for plants.

Lastly, they can be classified based on their (iii) active compounds such as organochlorines, organophosphates, carbamates, and pyrethroids [80,81]. Table 2 presents some pesticides according to their chemical classifications. Additionally, pesticides can be classified based on their route of administration and mechanism of action. Herbicides, fungicides, and insecticides account for more than 95% of agricultural pesticide use [81]. It is important to note that some pesticides, such as DDT, DDE, and DDD, have a half-time of 15 years [72].

In addition, some variations in classification are found in the world. According to the United States Environmental Protection Agency (EPA 2022), biopesticides fall into three main categories: microorganisms that control pests (microbial pesticides), naturally occurring substances that control pests (biochemical pesticides), and pesticidal substances produced by plants containing added genetic material or PIPs (plant-incorporated protectants) [82,83].

### 3.1. Challenges and Limitations of Biopesticides

Biopesticides are recognized for their effectiveness even in minimal quantities, and because they decompose quickly without leaving pollutants, they are a crucial element in integrated pest management. However, despite the many benefits already mentioned, several significant obstacles prevent their widespread adoption. The main one is the production cost, which is still high compared to synthetics. This cost, in addition to the production itself, also includes screening, development, and obtaining approval for regulations that are not yet fully established. Another two-way street point is the useful life of biopesticides, which are sensitive to variations in temperature and humidity, which can compromise their action. Farmers may need to purchase and learn how to use various agents to combat different pests, which can lead to confusion, high costs, and time consumption [84].

According to Šunjka et al. (2022), biopesticide is the main ecological solution to chemical agents; however, several factors must be improved for its wide acceptance. According to the authors, focus should be placed on developing new substances with optimization of formulation techniques. For example, neem is a widely used botanical biopesticide with environmental instability but is only temporarily effective. Other significant challenges are a limited spectrum of activity, unintended interactions with non-target organisms, variability in strain virulence, inconsistent product supply, sensitivity to environmental conditions (humidity and temperature), and exposure to light and UV radiation [85].

Research into the problem of action on non-target organisms is ongoing. Cappa et al. (2024) studied the harmful effects of the fungal biopesticide *Beauveria bassiana* on the wasp *Polistes dominula*. The study used an ecotoxicological approach that emphasizes an ecological perspective, going beyond assessing damage to isolated individuals, taking into account the health of colonies. As a result, they reported several harmful effects on *P. dominula*, such as altered locomotion, increased mortality, and colony failure. The study highlights how critical the ecotoxicological assessment of biopesticides is [86].

Dar et al. (2019) highlight other limitations, such as inadequate quality, inconsistent field results, unpredictable stability under field conditions, and high cost [87].

Despite the limitations that must be overcome with the advancement of research, several studies point to a promising future for biopesticides. For example, we can cite the management of potato common scab (CS), a disease caused predominantly by saprophytic bacteria of the Streptomyces genus. Coffin et al. (2020) studied the effectiveness of two Bacillus-based biopesticides (Double Nickel 55™ and Microflora PRO™) with the phosphite-based fungicide Phostrol™, independently and combined to combat CS. From the results presented, Bacillus biopesticides increased the number of suberized cell layers. The combination of Double Nickel 55™ and Phostrol™ is the most effective in consistently reducing the severity of CS [88].

Biopesticides have also been evaluated in the control of pecan weevil, which is an important pest that endangers *Carya illinoinensis* ([Wangenh.] K. Koch) (Fagales: Juglandaceae) orchards. Shapiro-Ilan et al. (2017) evaluated three agents based on the bacterium *Chromobacterium subtsugae* Martin, entomopathogenic nematode *Steinernema carpocapsae* (Weiser), and entomopathogenic fungus *Beauveria bassiana* (Balsamo) Vuillemin. These experiments were conducted over two consecutive years. According to the authors, the biopesticides presented satisfactory results, demonstrating that they are a viable and effective alternative to traditional chemical insecticides to control pecan weevil infestations in organic and conventional orchards [89].

*Spodoptera frugiperda*, known as fall armyworm, is a pest originating in the Americas; however, it has recently been found in Africa, posing risks to maize/cereal production. Aniwanou et al. (2021) investigated the effectiveness of bioproducts (biorational) and a semi-synthetic insecticide. In field trials involving maize cultivation, biorational insecticides produced comparable or better control than Emacot 19 EC (emamectin benzoate) [90].

Ataide et al. (2024) evaluated 21 conventional and 11 biorational insecticides against the invasive pest *Thrips parvispinus* (Thysanoptera: Thripidae). Each insecticide was tested on larvae and adults using direct exposure and residual toxicity. Of the biorational insecticides, mineral oil (3%) and sesame oil stood out, which caused greater mortality and less damage and more significant mortality against *T. parvispinus.* Finally, the authors suggest a rotation program between the insecticides evaluated to maximize the control of this pest [91].

### 3.2. Circular Economy and Production of Biopesticides

In addition to their environmental friendliness, bioproducts offer renewable and sustainable production and, importantly, significant economic advantages. This creates new business opportunities within the circular bioeconomy concept. Examples include using organic agro-waste and biorefinery practices to obtain biobased products.

Biowaste can be transformed into bioproducts or a mass of microorganisms used as biopesticides through fermentation [92]. Biological waste from agriculture, food processing, aquaculture, forest, pharmaceutical industry, agro residues such as babassu, other crops, and sewage sludge, among others, can be used. This practice reduces environmental pollution and decreases biopesticides and other bio-input production costs, allowing for the biotransformation of these residues into valuable products through fermentation on bioreactors [93]. This productive chain expresses the circular economy, a closed-loop resource management system [94].

There are several examples in the literature of biopesticides and circular economy. *Bacillus subtilis* RB14-CS under solid state fermentation (SSF) produces iturin A, an antifungal compound using soybean curd residue (okara) [95].

Using agro-industrial substrates such as rice bran and coconut husks, *Streptomyces similanensis* cells are produced through solid fermentation to protect orchids from root rot disease caused by *Phytophthora palmivora* [96].

A culture medium based on pearl millet as a substrate was utilized to produce *Akanthomyces lecanii* conidia for biopesticide application [97].

In this context, studies and developments in bioprospection, bioprocesses, solid-state and submerse fermentation optimization, scaling up, formulations, and strategies for field application are needed for more sustainable agriculture and commercialization of microbial pesticides [98].

## 4. Botanical Biopesticides—Phytopesticides

Botanical pesticides are considered advantageous alternatives to conventional chemical pesticides in agriculture due to their human and environmentally friendly nature. These pesticides are natural chemicals extracted from plants that act as repellants, attractants, antifeedants, and growth inhibitors. Plant essential oil from leaves, roots, stems, fruit seeds, fruit peels, wood, bark, resin, and flowers are rich in secondary metabolites such as terpenes, aldehydes, ketones, and phenolics that can be used as biopesticides, and their use as insecticides is growing [99].

Botanical pesticides are also utilized as nematicides, fungicides, bactericides, and virucides [100]. The development and use of botanical pesticides in agriculture still need to improve, indicating limitations, such as the vast quantum of raw materials required to produce plant-based pesticides. In addition, the variability of active compounds in different geographic areas is a bottleneck in standardization. The regulatory approval costs and short shelf-life are other problems [101]. Table 3 provides examples of how botanical ingredients can be used as biopesticides.

## 5. Microbial Biopesticides

The microorganisms belonging to fungal, bacterial, and viral groups represent a significant source of food loss, reaching up to 40% during production, postharvest storage, shipment, and marketing. Their growth and reproduction in the field are influenced by environmental variables such as temperature, humidity, and ultraviolet (UV) ray radiation [115]. The active ingredients of microbial pesticides primarily consist of microorganisms and their bioactive compounds. Microbials and their products acting as biopesticides are mass-produced by industrial fermentation and require multiple applications, as their populations typically sustain themselves for only a limited period [116]. The microorganisms have strong hydrophobicity and tend to form particles in solutions, which makes them challenging to use. To address this issue, adjuvants can be applied to deliver the active ingredients to their targets. Adjuvants enhance the physical and chemical properties of microbial pesticide formulations, improving their effectiveness [117].

## 6. Producing Microorganisms

Microbial biopesticides (MBPs) comprise naturally occurring microorganisms, including fungi, bacteria, baculoviruses, or nematode-associated microbes. They can be economical and present little or no toxicity to humans or non-target organisms. MBPs effectively tackle issues like insect resistance and target modification. More than 3000 distinct microbes that can infect and kill insects are now recognized [117,118].

MBPs have an efficiency comparable to chemical pesticides, with the advantage of lacking pathogenicity or toxicity to non-target microorganisms and macroorganisms, which can allow for their safe application close to harvest time. Its decomposition guarantees its non-persistence in agricultural products, maintaining the environment’s integrity such as groundwater and soil. The mechanism of action of microorganisms in biopesticides can occur through several mechanisms, such as the action of antimicrobial compounds, secretion of lytic enzymes, and competition with phytopathogens for essential resources such as nutrient absorption [119].

Unlike synthetic pesticides, MBPs have targeted action and can be obtained at affordable prices without the use of toxic and expensive chemical materials, offering an environmentally sustainable solution. However, MBPs also have limitations and disadvantages such as (a) high target specificity—not desired when aiming to control several pests simultaneously; (b) short shelf life—may be a disadvantage if the objective is to combat existing pests and prevent future infestations; and (c) low persistence in soil environments. Unlike synthetic pesticides, these are characteristics of a natural product [74]. Notably, 55% of biopesticides sold worldwide are of microbial origin, demonstrating their potential in the global market [120].

Only 0.1% of chemical pesticides specifically reach their target, while 99.5% reach the environment, resulting in considerable damage to non-target organisms and posing a danger to human health [121].

### 6.1. Bacteria

Studying highly diverse microbial ecosystems, such as carbon-rich organic soils and food spoilage environments, can help discover new biopesticides. Soils are complex environments with billions of organisms and wide diversity in bacteria. Although there is no consensus in the literature [122], a recent study contradicts the belief that most bacteria are non-cultivable, demonstrating that up to 70% of plant-associated microorganisms can be cultivated in various media [123,124].

The plant rhizosphere is a highly competitive environment where bacteria compete with other microbes, including plant pathogens, for nutrients and resources. Therefore, they are a favorable environment for isolating various microorganisms capable of producing different antimicrobial compounds [123].

The use of natural antagonistic microorganisms can be an alternative in the control of numerous plant pathogenic diseases. This control occurs through complex relationships often involving the production of anti-pathogenic compound products, competition for space and essential resources, and the triggering of host defense mechanisms [125].

According to Negi et al. (2023) [126], antagonistic microorganisms can be recognized as biological control agents and are essential in sustainable management practices to increase agricultural productivity. These microorganisms can be discovered in diverse habitats, such as soil, and in associations with various plants, nematodes, and insects. These authors studied the biodiversity in antagonistic microbes from different sources and related the antagonistic microorganism vs. target pest/pathogen. Our group re-processed these data and analyzed these relationships by grouping the microorganisms by the genera and phyla of antagonists (Figure 2).

The heatmap analysis reinforces the antagonistic versatility of the Bacillota and Pseudomonadota phyla through the number of occurrences and the diversity in target microorganisms. Regarding antagonistic bacteria, the genus Bacillus stands out despite the existence of other important genera, where several species have already demonstrated efficacy against a wide range of plant pathogens. Among them, *Bacillus thuringiensis* strains (Bt) are the most frequently used biopesticides in agriculture, medicine, and forestry [118,125,126].

A biological strategy for insect pests is the use of transgenic crops. Since 1996, genetically engineered crops that express δ endotoxins (Cry proteins) from *B. thuringiensis* have been increasingly adopted by farmers in numerous countries [129]. *Bacillus thuringiensis* is one of the most effective and used bioinsecticides for biologically controlling pests, accounting for 75–95% of the microbial biopesticide market. *Ectropis obliqua* (Geometridae, Lepidoptera) is a leaf-feeding and polyphagous pest that attacks many host plants, including tea, soybean, sesame, and flowers. In China, Lepidoptera leads to more than 60% loss in tea yield and quality. The population’s resistance to certain chemical insecticides is increasing [130]. Still, presently, the biological control of *E. obliqua* with *Bacillus thuringiens* is the preferred method [131].

Bt is a notable entomopathogen due to its production of larvicidal crystal toxins. These toxins are encoded by cry genes and produced during the bacterial sporulation phase, generally transported on plasmids inside the cell. Cry toxins form inclusion bodies known as “crystals”, which, when ingested by larvae, are solubilized by the alkaline pH of the larva’s midgut, releasing the protoxin. Finally, the protoxin is processed by digestive enzymes, transforming it into its active form that binds to several receptors on the intestinal epithelial surface, causing pore formation, gut paralysis, decreased food uptake, and death of the larva [132].

Spore-forming Bacillus species (e.g., *B. amyloliquefaciens*, *B. subtilis*, *B. thuringiensis*), as well as Streptomyces, are used as biocontrol agents to combat soil-borne plant pathogens, insect pests, or plant-parasitic nematodes. Freeze-dried spore-based products combined with additives and adjuvants stand out as they exhibit significant resilience under stressful conditions and are suitable for long-term storage and transportation [120].

Sporine was the first commercial bioinsecticides derived from Bt, developed in France in 1938 and used mainly to control flour moth infestations. The United States began commercial production of Bt in 1958. And in 1996, genetically modified insect-resistant crops, known as Bt crops, emerged, showing significant effectiveness in reducing dependence on chemical insecticides [133].

Bt subspecies produce a range of one to six diverse Cry toxins, this diversity being deterministic concerning their specific action against different phylogenetic orders. Despite this diversity, only four species were commercialized as bioinsecticides due to their high specificity against agricultural pests: (a) lepidopteran larvae: *B. thuringiensis* subsp. kurstaki and *B. thuringiensis* subsp. aizawai; (b) coleopterans: *B. thuringiensis* subsp. morrisoni; (c) mosquito larvae and dipterans: *B. thuringiensis* subsp. Israelensis [120].

Cry toxins are a diverse group of insecticidal proteins originating from Bacillus species, not belonging to a single homogeneous family of proteins. They encompass a wide range of unrelated lineages, classified into 78 different types. These toxins have well-documented toxicity against various targets, including Lepidoptera, Coleoptera, Hemiptera, Diptera, nematodes, parasites, some snails, and human cancer cells of different origins (Figure 3) [133,134,135].

Besides producing Cry proteins, BT synthesizes vegetable insecticidal protein (Vip) and secreted insecticide protein (SIP) during its vegetative growth phase. Vip is distinguished by unique sequence homology and different receptor sites from Cry proteins and is considered an excellent toxic candidate. Referred to as second-generation insecticidal proteins, Vip proteins can be utilized alone or combined with Cry proteins to manage various pests effectively [135,136].

Combining Cry and Vip proteins may exhibit synergistic insecticidal activity against a broad spectrum of insect pests. Hemthanon et al. (2023) isolated Bacillus thuringiensis that showed high production levels of Vip3A and Cry proteins and evaluated thermostability to control Spodoptera spp. In the work, they demonstrated that the Bt 506 isolate produced proteins identified as Cry1Aa45, Cry2Aa22, and Vip3A87, and evaluated the insecticidal potential of crop extracts. As a result, the extract showed high toxicity against second-instar larvae of *Spodoptera exigua* and *Spodoptera frugiperda*, maintaining activity even after thermal stress (50 °C) [137].

Konecka et al. (2020) studied the synergistic action of the combination of the botanical compound carvacrol with the crystalline Cry proteins from Bt targeting *Cydia pomonella* L. (Lepidoptera: Tortricidae) and *Spodoptera exigua* Hübner (Lepidoptera: Noctuidae). The results demonstrated that this combination at specific concentrations (carvacrol 0.1%, Cry proteins 0.05%) synergistically reduces L1 and L3 larval populations. Therefore, the authors suggest that this synergy has the potential to be used against lepidopteran pests [138].

Spinosyns are secondary metabolites produced from the aerobic fermentation of the Gram-positive filamentous bacterium *Saccharopolyspora spinosa*. Initially, strains producing spinosad were selected, which consists of a mixture of the two most active compounds: spinosyns A and D [139,140].

Spinosad (SPI) is now a commercial bioinsecticide registered in more than 30 countries to control pests and was first commercialized in 1997 by Dow AgroSciences. SPI is a broad-spectrum bioinsecticide that is effective against lepidopterans, such as *Helicoverpa armigera*, *Plutella xylostella*, *Spodoptera exigua*, and *Heliothis virescens* [139,141].

SPI targets the insect’s nicotine acetylcholinesterase receptors (nAChRs), which, upon binding, acts as an allosteric modulator, disrupting the nervous system’s GABA-gated ion channels—resulting in paralysis and death of the insect [141,142].

Kljajić et al. (2023) analyzed the residual effects of insecticides (organophosphate malathion, spinosad, pyrethroid deltamethrin, and piperonyl butoxide [PBO]), monitoring 210, 265, 310, and 365 days after treatment. The authors investigated the insecticidal effect on weevil survival using two bean varieties. As a result, they indicated that the insect’s survival depends on the duration of exposure and the subsequent recovery period, and there was a slight variation in the target survival rate for the different insecticides. Briefly, spinosad showed substantial long-term protection against *A. obtectus*, whereas malathion had limited performance [143].

### 6.2. Virus

Various entomopathogenic viruses attack and kill many insects. Few are commonly found in insect populations, including Baculovirus, Cypovirus, Entomopoxvirus, and Iridovirus. Some insects are crop pests and particularly susceptible to these viral infections, making these viruses valuable as biological control agents [144].

Baculoviruses are dsDNA viruses classified within the Baculoviridae family into two groups, nuclear polyhedroviruses (NPVs) and granuloviruses (GVs), both requiring ingestion by the insect to initiate infection despite their effectiveness as biological control agents (BCAs) [145]. They are the most common and effective type of insect virus, known to infect over 600 insect species worldwide [144]. As a natural host, this family infests some orders, such as Lepidoptera, Hymenoptera, and Diptera. They are pathogens that exploit insects’ feeding behaviors for transmission. They spread through contaminated food sources, making them adequate for controlling some crop pests [74,144].

Baculovirus-infected insects become whitish because of the massive infection of the fat body, visible through a more translucent integument (exoskeleton), which turns thinner as the disease advances until it ruptures. The viruses are embedded within a proteinaceous body named occlusion bodies (OBs). After the occlusion, bodies dissolve in the insect’s midgut, releasing viral particles that invade midgut epithelium and other tissues, causing extensive damage and eventually leading to the insect’s death [146].

Baculoviruses have been used in crop protection as commercial biological insecticides since the 1980s. They are a prominent example of viruses used as biological control agents. These entomopathogenic viruses are managed by various agricultural and forest insect pest populations worldwide as biopesticides [147,148].

In addition, entomopathogenic viruses are highly host-specific, making them excellent candidates for prospecting new biopesticides [149,150]. Historically, commercial use of baculoviruses faced challenges such as high costs, limited efficiency, narrow host ranges, and susceptibility to environmental factors like UV light and field stability [151]. Despite these challenges, virus-based BCA remains integral to integrated pest management (IPM) strategies, mainly due to their role in sustainable pest control practices without significant environmental impacts on non-target organisms [144,152].

Unquestionably, many more insect viruses still need to be discovered; however, only a few have demonstrated potential as control agents, particularly baculoviruses [144].

Approximately 60 baculovirus-based insecticides have been commercially developed to manage diverse insect pests globally [144,153,154].

Continued research into their genetic manipulation, formulation strategies, and ecological impacts holds promise for further enhancing their efficacy and integrating them more extensively into modern agricultural practices [74,152]. Table 4 presents examples of commercially available baculovirus biopesticides registered for insect pest control in various countries.

### 6.3. Microalgae

Due to their biological activity, Microalgae and cyanobacteria produce different metabolites that can be used in agriculture as biofertilizers, biostimulants, or biopesticides. Cyanobacteria act as primary biological pest control agents, helping to manage various plant diseases. They produce bioactive compounds that combat fungi, nematodes, and other diseases by inactivating essential enzymes, disrupting cytoplasmic membranes, and inhibiting protein synthesis.

Additionally, they stimulate the plants’ immune systems and enhance the production of antioxidant compounds for disease resistance. The secondary metabolites produced by the microalgae *Anabaena laxa*, *Calothrix* sp. *Chlorella vulgaris*, and *Fischerella* sp. show antimicrobial, larvicidal, and insecticidal activities [175,176,177,178].

Chlorellin is a mixture of fatty acids that was the first antibacterial biopesticide produced by the green algae Chlorella. This compound inhibits Gram-positive and Gram-negative bacteria [179]. Various compounds in microalgae have been used against phytopathogens—for example, antibacterial alkaloids from *Fischerella* sp. [180]. Nostofungicine is a lipopeptide with fungicide activity isolated from *Nostoc* sp. [181]. Additionally, eicosapentaenoic acid (EPA) is an antimicrobial fatty acid found in the diatom *Phaeodactylum tricornutum* [182]. Cyanobacteria compounds also function as herbicides. A phenolic substance called cyanobacterin, isolated from the Cyanobacteria *Scytonema hofmanni*, is a potent inhibitor of photosynthetic electron transport in photosystem II [183,184].

Some substances from microalgae are considered allelochemicals and have insecticidal properties. For instance, extracts from *Chlamydomonas reinhardtii*, combined with microparticles of zinc oxide, doubled the mortality of mealworm beetle larvae (Tenebrio molitor) compared with using zinc oxide only. In addition, the acetonic extract from the diatom *Amphora coffeaeformis* and the Chlorophyta *Scenedesmus obliquus* showed larvicidal activity against the mosquito *Culex pipiens*. Phenolic compounds such as benzoic acids (gallic acid, methyl gallate, and syringic acid), cinnamic acids (caffeic acid and cinnamic acids), and flavonoids (taxifolin, kaempferol, and naringenin) were identified in these extracts [185].

Microalgae have the potential to be a source of innovative biopesticides, and this field needs to be explored to promote further development. It is important to note that the safety and environmental impact of using these metabolites as biopesticides are also crucial considerations. Research in this area is ongoing to ensure their use is safe and sustainable, aligning with sustainable agriculture principles.

### 6.4. Fungal Biopesticides

Microorganisms can synthesize secondary metabolites with diverse chemical structures and biological activities [76]. Newly discovered antifungal compounds are more likely to exhibit novel and more specific modes of action than commercial fungicides [83]. Regarding microbial biopesticides, fungi play an important role in the market share and serve as an eco-friendly substitute for hazardous agrochemicals harmful to humans and the environment. The *Trichoderma* genus is among agriculture’s most utilized fungi as a microbial-based pesticide [186]. Table 5 exemplifies some advantages and disadvantages of fungi-based pesticides [187,188].

Fungi have developed the capacity to produce various secondary metabolites as an adaptative strategy to survive in challenging environments. Biocontrol strategies in crops rely on these bioactive compounds and their antibiosis mechanism against crop diseases, particularly those with antimicrobial properties [83,189].

Xu et al. (2021) [190] identified 132 antifungal metabolites isolated from fungal endophytes in the past two decades. Some of those metabolites are presented in Table 6, along with the control of some phytopathogens [83].

While these compounds are naturally abundant, they increasingly serve as the foundation for synthetic fungicides. Moreover, given the broad spectrum of chemical families these metabolites belong to, they can trigger plant resistance, prompting the production of metabolites that enhance natural protection [83]. In addition to the fungal pathogen targets listed in Table 6, fungal-based biopesticides can also serve as insecticides [197], nematicides [198], and herbicides [199].

#### 6.4.1. Entomopathogenic Fungi (EFP)

A unique class of fungi biopesticides is the entomopathogenic fungi. They have a wide range of coverage for various types of insects, and this is an important advantage: lack of specificity. Another differentiated point of EFP in pest biocontrol is their persistence in the host’s environment for the required infection period. Environmental competence is a desirable factor contributing to this successful approach [200,201].

The genera *Metarhizium* and *Beauveria*, which are soil-inhabiting fungi, are emerging as agents for biocontrol of lepidopteran pests in agriculture [202]. Other species, including *Paecilomyces farinosus* and *Verticillium lecanii*, are significative biocontrol agents [198,203]. In contrast, they have limited resistance development to EFP use as biopesticides. Lepidopteran insects resist chemical pesticides and adapt using genetic and physiological mechanisms [204]. These fungi employ various strategies to penetrate the insect cuticle and initiate infection, including the secretion of hydrolases such as peptidases, chitinases, esterases, and lipases. It is a complex process involving spore recognition, adhesion, germination, and differentiation into infective structures [202]. *Metarhizium anisopliae* can infect over 200 insect pests of major plants belonging to more than 14 insect orders, including Acari, Malacostraca (amphipods), Dermaptera, and Ephemeroptera, which are known to infect species belonging to the superfamily Acridoidea [205]. *Beauveria* species also exhibit a broad host range and parasitize at least 15 different insect orders, including Blattariae (cockroaches), Coleoptera (beetles), Diptera (flies), Dermaptera (earwigs), Embioptera (web spinners), Hymenoptera (bees, wasps, and ants), Hemiptera (true bugs in the suborders Auchenorrhyncha, Coleorrhyncha, Heteroptera, and Sternorrhyncha), Isoptera (termites), and Lepidoptera (moths and butterflies), as well as arthropod species belonging to the classes Gastropoda and Arachnida [202]. *Beauveria bassiana* and *B. brongniartii* are well-known environmentally safe alternatives to using chemical pesticides to control agricultural pests. Products based on *Beauveria bassiana* have been developed in many countries worldwide [206]. Table 7 shows some examples.

#### 6.4.2. Mechanisms of Action

The mechanisms of action against fungal pathogens may differ depending on the fungal endophyte and its interaction with the host. Usually, four distinct mechanisms are considered: (i) competition, (ii) antibiosis, (iii) mycoparasitism, and (iv) induction of plant resistance [83]. Entomopathogenic fungi produce spores known as conidia, which adhere to and penetrate the cuticle of arthropods, initiating a lethal infection and regulating their natural population [83,221].

The first mechanism, competition, is an indirect form of antagonism employed by biocontrol agents. It does not entail direct interaction between the pathogen and the antagonistic microbe. Instead, they compete for nutrients, space, essential micronutrients like iron and manganese, and specific growth substances or germination stimulants. Biological control agents (BCAs) produce high-affinity iron chelators called siderophores, which deprive pathogenic microbes of essential iron, starving them of a crucial element [222].

A biological control agent can also employ a direct mode of action characterized by the secretion of a diverse array of secondary metabolites. These compounds have various roles: antibiotics, effectors, elicitors, and degrading enzymes. This mechanism is commonly referred to as antibiosis. Numerous microorganisms, including a significant proportion of known BCAs, produce and release secondary metabolites with antimicrobial properties as part of their ongoing competition with other microorganisms in their natural habitat [223].

The third mechanism is mycoparasitism, a biological control mechanism employed by *Trichoderma*. It involves a complex process to identify and neutralize phytopathogens [83,224,225]. This process unfolds through several stages: initial recognition of the target organism, followed by chemotrophic growth toward it, leading to a multifaceted attack combining physical and chemical means to eradicate the threat while utilizing nutrients [226]. *Trichoderma*’s detection of phytopathogens involves recognizing lectins binding to carbohydrates in the pathogen’s cell walls. Subsequently, *Trichoderma* hyphae envelop the pathogen’s hyphae and secrete cell wall degrading enzymes (CWDEs) like peptidases, glucanases, and chitinases to dismantle the pathogen’s cellular components [227,228]. These enzymatic actions create openings facilitating *Trichoderma* sp. penetration into the pathogen’s interior, enabling parasitization while disrupting crucial enzymes necessary for the pathogen’s colonization of plant tissues [229].

This mycoparasitic mode of action exemplifies direct antagonism, characterized by the inhibition or destruction of organisms by producing metabolites by fungal biocontrol agents [230]. These metabolites, including volatile or non-volatile toxic molecules like antimicrobial substances or lytic enzymes, are synthesized by fungi in response to substrate depletion, preventing other microorganisms from competing for resources and colonizing the same niche [115].

The last mechanism is plant resistance, which encompasses the host’s ability to delay or prevent pathogen colonization and development through various defense mechanisms, categorized as biochemical or structural and temporally as preformed or postformed [231]. Induced resistance involves the enhancement of constitutive and inducible defenses in plants through biotic or abiotic stimuli, resulting in increased defense capacity against pathogens and pests. This phenomenon offers potential for controlling plant diseases and serves as a tool for studying resistance mechanisms and plant susceptibility to phytopathogens [232,233]. It boasts several advantages, including effectiveness against various pathogens, abiotic stresses, and stability due to the collaboration of different resistance mechanisms [233]. The desired outcome of induced resistance is the “priming” state, where plants are in a heightened state of alertness, ready to express resistance mechanisms more intensely upon encountering stressors, without significant energy expenditure [83,223,234]. This mechanism of action can take two forms: localized protection limited to the treated tissues and systemic resistance extending beyond the application site. This activation of plant defense mechanisms can result from various compounds found in plant extracts, yeast preparations, growth-promoting microorganisms, avirulent pathogens, and endophytic fungi [235,236,237].

Various mechanisms for inducing the host plant defense system have been documented. For example, the endophytic fungus *Serendipita indica* (formerly *Piriformospora indica*) enhances resistance in barley and rice against different pathogens [238].

In barley, induced plants developed structures that hindered the penetration of *Blumeria graminis*, while in rice, induced plants increased their antioxidant capacity to limit pathogen severity. *Trichoderma harzianum* also induces fungal pathogen resistance by enhancing lignification enzymes in crops like maize and strengthening epidermal cell walls in others such as cucumber [234]. Those responses are mediated by salicylic acid signaling pathways, leading to the production of pathogenesis-related proteins and oxidative enzymes [83,239]. The specificity of microbial pesticides presents both advantages and challenges. While it diminishes the risk of toxicity to humans, the environment, and other beneficial organisms in agricultural settings, it also poses difficulties for farmers to apply it in practice [240,241].

### 6.5. Entomopathogenic Nematodes and Their Bacterial Symbionts

Entomopathogenic nematodes (EPNs) are soil-inhabiting organisms effective in controlling insect pests, including caterpillars, cutworms, crown borers, corn rootworms, crane flies, thrips, and beetles. Commercial successes of RPNs use are documented for Diaprepes root weevil, black vine weevil, fungus gnats, thrips, certain borers, white grubs, billbugs, mole crickets, and sciarid flies. These pests cause crop losses. In addition, the nematode *Phasmarhabditis hermaphrodite* is commercialized against slugs [242,243,244]. There are 23 nematode families known to have anti-insect properties, Steinernematidae and Heterorhabditidae being the most important. *Steinernema* sp. and *Heterorhabditis* sp. are associated with symbiotic bacteria: *Xenorhabdus* spp. and *Photorhabdus* spp., respectively. These bacteria cause septicemia that leads to the death of insects within 24–48 h. The symbionts spread and multiply in the hemolymph of the insect pest and kill them by septicemia. Toxins and virulence factors produced by the bacterial symbiont also contribute to evading the host immune response by degrading hemocytes and phenoloxidases [245]. Furthermore, bacteria alone cannot penetrate the gut and cannot independently gain entry to the host’s hemocoel [246].

The surface coat proteins of entomopathogenic nematodes are also related to suppressing the host immune response, which protects the nematode from being killed [247]. EPNs are easy to apply and are effective against most lepidopterans. They are easy to cultivate can recycle plant nutrients persisting in the environment, and are safe for non-target organisms [244].

### 6.6. Microsporidia

Microsporidia are eukaryotic, with vast diversity, single-celled, spore-forming, intracellular parasites with an extensive host range, such as honeybees, silkworms, and other insects. They also can parasite fish, rodents, rabbits, primates, and humans [248,249,250].

Although once thought to be protists, they are considered to be related to fungi. Currently, Microsporidia are divided into six groups: Glugeida (previously clades 5 and 3), Nosematida (previously clade 4a), Enterocytozoonida (previously clade 4b), Amblyosporida (previously clades 3 and 5), Neopereziida (previously clade 1), and Ovavesiculida (previously clade 2), and an additional unnamed group containing *Hamiltosporidium*, *Neoflabelliforma*, *Areospora*, and *Astathelohania* species. The ribosomal small RNA subunit (SSU) analysis cannot robustly resolve the deep phylogenetic relationships of microsporidian clades. Complete resolution of the phylogenetic structure of Microsporidia requires larger datasets, which are increasingly available from whole genome sequencing studies. The classification issue will be clarified with future genetic data [251].

Microsporidia naturally infect a wide range of insects and act relatively slowly, taking days or weeks to debilitate their hosts. In addition, their ability to weaken insects makes them more vulnerable to adverse weather conditions and other biotic factors, such as predators and parasitoids, which can enhance their overall impact. They reduce host reproduction or feeding rather than killing the pest outright [252].

The insect is infected by ingestion. However, some natural transmission within pest populations is also via predators and parasitoids. The pathogen enters the insect body through the gut wall. After this, it spreads to tissues and organs, multiplies, and can cause tissue breakdown and septicemia, leading to death if the infection rate is high [253].

Microsporidia are potentially used as a biological control agent for insects impacting crops, presenting a promising solution for pest management [254]. Some entomopathogenic species:*Nosema pyrausta* (*Perezia pyraustae*): This microsporidium infects several insect species and can be an important biological control agent. It effectively helps regulate the corn borer, *Ostrinia nubilalis.* However, its commercial use is still in the early stages [255,256].*Nosema locustae*: The only commercially available species of microsporidium has a vast host range, infecting 121 orthopteran species. This microorganism has been used successfully to control grasshopper populations. *N. locustae* is a promising biopesticide for locust control due to its potential for ultralow-volume production and dissemination [257]. The use of microsporidia *Nosema locustae* and *Paranosema locustae*, *in combination* with *the fungi Metarhizium anisopliae* var. *acridum*, were found to be the best biological control agents (BCA) [258,259].*Vairimorpha necatrix:* This microsporidium has commercial potential in a broad host range among caterpillar pests, including corn earworms. It can be more virulent than other species, with infected insects potentially dying within six days of infection [260]. It was tested as a microbial insecticide for the tobacco budworm *Heliothis virescens*. It can be more virulent than other species, with infected insects potentially dying within six days of infection [260].

Species belonging to the genus Nosema or Vairimorpha were found infecting *Gonipterus platensis* (Coleoptera: Curculionidae), a major destructive pest of eucalyptus plantations in Brazil. The symptoms of the infected Coleoptera were identified by an abnormal abdomen and malformation of the second pair of wings, impairing their flight activity. The finding is promising for elaborating a biopesticide for this crop pest [261].

### 6.7. Apicomplexa

Eugregarines, the most diverse order of gregarines, with 240 genera, frequently infect insects and are the most common protozoa in scarab species. *Stictospora* species infect grubs such as Japanese beetle, Zealand grass grub, Oryctes, Rhizotrogus, and Melolontha. *Mattesia* species and *Aranciocystis mukarensis* were found infecting cereal chafer [262]. Vaselek (2024) identified several gregarine species and their sandfly host species, with *Ascogregarina chagasi*, later named *Psychodiella chagasi*, being the most studied sandfly gregarine [263].

## 7. Biopesticides Specificity

Depending on the conditions and mode of action of fungi, they exhibit different degrees of specificity. Microbial biocides, such as those based on *Bacillus thuringiensis*, are known for their specificity against target pests, minimizing impact on beneficial organisms and the environment. It is essential to note that the term specificity used for *Bacillus thuringiensis* englobes different groups of insects of Lepidoptera order [264].

Entomopathogenic fungi are effective against various insects, providing the critical advantage of minimal resistance development [265].

In some cases, the degree of specificity correlates with nutritional requirements. Species with complex dietary requirements tend to have narrow host ranges [265].

The use of various microorganisms as biocontrol agents for pests demonstrates specificity against the target disease and pathogen, often necessitating the application of multiple microbial pesticides [266]. The combination of biopesticides is known as a biosolution. For example, pyrethrum is a highly effective botanical bioinsecticide with low toxicity to mammals. However, it can be degraded by ultraviolet (UV) light. On the other hand, the entomopathogenic fungus *Beauveria bassiana* is more persistent than pyrethrum, but it takes several days to cause insect mortality. Promising results have been obtained by combining pyrethrum with *Beauveria bassiana* to control *Spodoptera frugiperda*. Depending on the concentration, an additive effect was observed [267].

## 8. Delivery System of Biological Control Agents after Production

Advancements in application technology have revolutionized agricultural practices, offering solutions to diverse challenges. These technologies encompass a spectrum of tools and equipment designed to optimize the delivery of farming inputs, including biological control agents (BCAs), pesticides, fertilizers, and other essential substances [268,269]. BCAs are applied to plants through various methods, such as seed inoculation, soil application, or foliar spraying, depending on factors like their mode of action and the plant’s growth stage [270,271].

These innovations in application technology facilitate precise, efficient, and sustainable management practices in agriculture. These advancements, from precision spraying systems to intelligent irrigation technologies, help farmers achieve long-term agricultural production sustainability [76,272].

### 8.1. Soil Application

Soil application is an effective method for delivering biological control agents against soil-borne pathogens, involving the direct application of the formulation into the soil, typically within the seeding furrow. This technique employs various forms of inoculant application, including granular inoculants, wettable powders, water-dispersible granules, and liquid inoculants [76,271].

Granular applicators commonly apply these formulations beneath, above, or alongside the seeds. Although powder inoculants are suitable for soil application, they are dustier and less user-friendly than granular forms [273,274]. Wettable powders, water-dispersible granules, and liquid formulations can be evenly distributed over crop areas by hand or with mechanical spraying equipment. Additionally, these formulations can be applied directly to the root zone of individual plants through drip irrigation or hydroponic systems [275]. Soil application is particularly advantageous when introducing large populations of BCAs into the soil because it avoids damage to fragile seeds and mitigates the adverse effects of seed-applied chemicals [271,276].

### 8.2. Seed Treatment

To optimally protect germinating seeds and seedlings against disease, biofungicides must be applied to colonize the spermosphere and rhizosphere at densities high enough to suppress pathogens [271]. Biocontrol agents can be used through precoating, encapsulation, in-furrow applications, or incorporation into soil mixes. Precoating involves dry powders or oil- and polymer-based liquids with dormant microbes, often enhanced with additives for better adhesion [76,270]. Seed coatings must consider inoculum density, coating stability, and production feasibility. Applying formulations at planting ensures a high number of viable microbes but relies on correct application by growers [277].

Techniques like seed priming, which controls seed hydration and pre-germinative activities, also play a crucial role [278].

### 8.3. Spraying Techniques

Formulating bacteria or fungi for foliar sprays involves tailoring the formulation to suit the specific crop, targeted pests, and the chosen delivery system. Among the standard formulations used are liquids and slurries, each offering distinct advantages regarding ease of application and microbial viability [76,279]. Additives like emulsifiers, stickers, spreaders, and other adjuvants play crucial roles in facilitating application, promoting dispersal, enhancing adhesion to plant surfaces, and protecting the microbes from adverse environmental conditions such as desiccation, unfavorable pH and UV radiation [280,281].

The successful application of foliar biopesticides hinges on several factors related to the physical properties of the spray suspension. Surface tension and viscosity are essential in reducing droplet size, ensuring proper dispersion, and maintaining control over droplet movement [282,283].

Different sprayer types, like hydraulic, air-blast, and electrostatic sprayers, are used to apply liquid pesticide formulations to crops [284,285].

The choice of sprayer directly affects droplet size and coverage, which is crucial for effective pest control. Timing is vital in foliar biopesticide application, with early morning or late afternoon preferred to minimize UV exposure and desiccation stress on applied microbes, enhancing their effectiveness [286,287].

Successful foliar biopesticide application involves selecting appropriate formulations, considering spray properties, using suitable adjuvants, precise application techniques, and strategic timing to maximize biocontrol agent efficacy on plant surfaces [76,281].

### 8.4. Endotherapy

Trunk injection, also known as “endotherapy”, involves directly applying crop protection materials into the stems or trunks of woody plants, offering an alternative to traditional methods. It consists of injecting substances into the tree’s xylem, where they are distributed throughout the plant via the transpiration stream [288,289].

Endotherapy has some benefits when compared to conventional methods: (1) more precise and efficient application of products; (2) elimination of spray drift; (3) reduced risk of worker exposure when applied correctly; and (4) minimized impact on non-target organisms [290,291].

Trunk injection has successfully controlled various pests and diseases, including fungal and bacterial pathogens; chewing, sap-sucking, and wood-boring insects; and nematodes [292].

Despite its widespread use in agriculture, biocontrol is still seldom applied in forestry, with most efforts confined to laboratory, greenhouse, or nursery settings. Forestry poses specific challenges for biocontrol application by conventional methods, such as steep terrain, lack of access roads, and the large size of trees [293,294]. However, endotherapy can address some of these issues by preventing the environmental dispersal of biological control agents and reducing the doses needed for treatment. Despite the restricted use for forestry, the effectiveness of endotherapy, combined with *Trichoderma* spp., has been demonstrated [295].

## 9. Adjuvants Used for Microbial Pesticide

Adjuvants can increase the effectiveness of biopesticides due to their ability to improve the physical and chemical properties of preparations, increasing the likelihood of more effective pest control. As it is applied with biopesticide, it is essential to investigate the impact of different types of adjuvants on non-target microorganisms and the environment [280,281].

Most adjuvants currently used in microbial biopesticides are still the same as those developed for chemical pesticides. Considering the evolution of biopesticides, it is important to jointly develop “green adjuvants” that are equally environmentally friendly [281].

Some types of adjuvants have already been used for the production of biopesticides of microbiological origin: surfactants, carriers, protective agents, nutrients, phagostimulants (feeding stimulants), pH Buffers, and UV-protective agents [280,296,297,298].

### 9.1. Surfactants and Carriers

The operation of the combination of protective and adjuvant substances is synergistic as it improves the survival of the microorganisms and facilitates mixing, application, and handling, respectively [299]. Surfactants enhance the interfacial tension between liquid pesticides and crop surfaces, thus increasing the permeability and wettability of formulations. It also features functions such as enhancing emulsifying, dispersing, spreading, and sticking.

Below are some examples of surfactants used in the production of microbial pesticides:

(a) Cationic surfactants: hexadecyl trimethyl ammonium bromide (TTAB); (b) anionic surfactants: sodium 2-butyl-1-naphthalenesulfonate (BX); (c) nonionic surfactants: Tween-80; (d) amphoteric surfactants: dodecyl dimethyl betaine (BS-12) [281,296].

Carriers are inert components that help deliver formulation to the target and may contain loaded or diluted pesticides. They are also crucial in promptly controlling the release of the active ingredient. Examples of carriers are alginate, carrageenan, and molasses [280,281,296].

Zhang et al. (2023) [298] analyzed a microbial inhibitor to control diseases and increased the growth of oats (*Avena sativa*). The authors used experimental design to screen and identify ideal concentrations of wetting agents, preservatives, and protective agents. As a result, they found that the best composition was Tween-80 (14.96 mL/L), methylparaben (5.12 g/L), and sorbitol (5.6 g/L).

Little is known about the behavior of biopesticides combined with adjuvants in organic onion production. That is why Iglesias et al. (2021) [300] studied three adjuvants (pinene polymers, potassium salts of fatty acids, and neem oil) and four bioinsecticides (*Isaria fumosorosea* Apopka strain 97, azadirachtin + pyrethrins, azadirachtin, and spinosad) in field trials. Spinosad and the combination with either neem oil or salts of fatty acids are effective in reducing thrips densities.

Bayramoglu et al. (2023) developed a novel betabaculovirus (HycuGV-Hc1) as a biopesticide (HycuGV-TR61) to target fall webworm, *Hyphantria cunea* Drury (Lepidoptera: Erebidae). The viral suspension was extracted from infected larvae and formulated with sunflower oil and various adjuvants, always following the rule of 20% active ingredient and 80% other fillers and adjuvants. The HycuGV-TR61 formulation achieved a kill rate of 99.86% at the highest concentration, demonstrating significant stability across various storage conditions. The study highlighted the success of a formulation with local viruses, the resilience of the formulation to environmental factors, and its promising application in pest control [301].

Kim et al. (2010) analyzed the roles of adjuvants, specifically corn oil and polyoxyethylene-(3)-isotridecyl ether (TDE-3), in increasing the aphicidal activity of enzymes in the supernatant of *Beauveria bassiana* SFB-205. As a result, they demonstrated that combining the enzyme extract with corn oil and TDE-3 achieved a control efficacy of 96.3% against cotton aphids, *Aphis gossypii* Glover (Hemiptera: Aphididae). This result surpassed the water-based formulation and the TDE-3 alone, which showed less than 20% effectiveness. The superior performance was attributed to corn oil’s ability to delay the formulation’s evaporation, resulting in more extended enzyme activity and the action of TDE-3 in breaking down the formulation chitin–protein matrix, facilitating greater penetration of the enzyme [302].

### 9.2. Protective Agents

Formulating and applying microbial pesticides in the field requires careful consideration of their inherent differences compared to chemical pesticides. The clearest among these disparities is that microbial pesticides exist as live particulate suspensions with a slower mode of action, while chemical pesticides are fast-acting solutions [280]. For instance, when applying microbial-based pesticides via foliar spray, environmental factors can adversely affect the survival rates of microorganisms, potentially leading to decay. In order to reduce this, formulations frequently include protective agents such as stabilizers and UV protectants [303,304]. The suboptimal performance of microbial pesticides in field trials has been associated with sunlight exposure. Efforts to bolster their effectiveness have predominantly hinged on applying UV-protectant adjuvants to enhance their survival rates [305].

Formulations with UV protectors shield against photodamage by two mechanisms: (i) absorbing, blocking, or reflecting harmful wavelengths, thus preventing them from reaching the cell surface of microbial-based pesticides, and (ii) scavenging reactive oxygen species (ROS) using antioxidant compounds [306].

Several compounds serve as effective UV protective agents; these include pectin/starch [307], as well as organosilicon surfactants [308], chitosan and chitosan-based nanoparticles [309], soluble starch dextrin, and casein [306].

The selection of UV protective agents depends on several factors, including the specific microbial species used in the pesticide formulation, the environmental conditions of the target application site, and the desired duration of protection [310].

## 10. Production of Biological Control Agents

Biomass production is the critical first step in formulating biological control agents. Culturing conditions determine population densities at harvest and influence the viability and fitness of microbes during formulation, storage, and application [311]. These conditions are specific to each microbial strain and must be carefully optimized to improve field performance [312]. Standard laboratory culture media, rich in peptone, yeast extract, and tryptone, enable high biomass production of BCAs on a small scale. However, their use for large-scale production is neither economical nor suitable [313]. Combining traditional components with low-cost substrates can reduce fermentation costs, as demonstrated for some BCAs [314]. Industrial byproducts such as cheese whey, corn steep liquor, soybean bran, wheat bran, barley fiber, fishmeal, molasses, composts, and wastewater sludge can serve as alternative carbon and nitrogen sources [315,316]. For instance, composts and wastewater derivatives have been used for fermenting *Bacillus amyloliquefaciens* and *Paenibacillus polymyxa* [314,317]. Optimal culture parameters, such as dissolved oxygen, temperature, pH, and mixing speed, are also critical [318]. In liquid fermentation systems, dissolved oxygen levels typically range between 90% and 100%, with temperatures maintained at 25–28 °C for BCA production [319,320,321].

Hence, fermentation methods are essential for BCAs. Inoculum production has two primary methods: solid-state fermentation and submerged (liquid-state) fermentation. Biphasic fermentation is used for some biopesticides [76]. These methods are discussed below.

### 10.1. Solid-State Fermentation (SSF)

In SSF, microorganisms grow on the surface of a solid substrate without free-flowing water. This method is advantageous for producing fungal BCAs and some bacterial spores [322]. The choice of substrate is crucial and often includes agricultural byproducts like rice bran, wheat bran, or sawdust [323,324].

Key parameters include moisture content, temperature, and aeration; for SSF, heat dissipation and maintaining aerobic conditions will maximize sporulation and overall yield, which is critical for the formulation of biopesticides [312,323,325].

### 10.2. Submerged Fermentation (SmF)

In SmF, microorganisms grow in a liquid nutrient medium inside a bioreactor, allowing for easy control of environmental parameters such as pH, temperature, and oxygen levels [326,327].

SmF offers advantages in scaling up production and achieving consistent product quality and reliability. The process requires careful monitoring and control of parameters involved in BCA growth. These parameters can be altered to enhance and optimize microbial growth and metabolite production. Several types of bioreactors are on the market, each presenting advantages and disadvantages. The bioreactor design must address economic and microbial requirements [328,329].

### 10.3. Biphasic Fermentation

Some biopesticide production processes benefit from combining SSF and SmF. For example, initial biomass production may occur in submerged culture, followed by sporulation or secondary metabolite production in solid-state culture. This hybrid approach can optimize yield and efficacy [313,330,331].

## 11. Formulation of Biological Agents

### 11.1. Formulation

Biomass production is the initial and foremost stage in formulating biological control agents. The conditions under which these organisms are cultured determine their population densities at harvest and significantly impact their viability throughout the formulation, storage, and application processes [241].

While standard laboratory culture media, enriched with peptone, yeast extract, and tryptone, are effective for generating high biomass in BCAs on a small scale, they are not cost-effective or suitable for large-scale production [268]. A blend of traditional components with low-cost substrates has been adopted to address this challenge, reducing fermentation costs. Some examples are industrial byproducts such as cheese whey [332], corn steep liquor [333], soybean bran [322], wheat bran [319], barley fiber [334], and molasses [335].

Biopesticide formulation involves blending it with inert materials, known as adjuvants, to confer specific advantageous qualities, thus creating its distinctive formulation. These adjuvants can be inert substances like talc, silica, and diatomaceous earth, alongside the addition of wetting agents, spreaders, emulsifiers, stickers, and stabilizers, all geared towards bestowing unique properties [269,336,337].

Using various adjuvants—ranging from polymers to emulsifying agents, surfactants, solvents, stabilizers, and defoamers—within formulation technology ensures the production of safe and effective products. These products guarantee ease of use and significantly enhance efficacy [307,309,310]:4.Pesticide formulation plays a pivotal role in optimizing storage, handling, safety measures, application methods, and the overall effectiveness of biopesticides [305,306];5.Moreover, formulation contributes to precise adherence to target sites and controlled release of bioactive compounds, tailoring its mechanism according to the specific formulation type [268,338];6.Formulation technology serves as a catalyst for elevating the quality standards of existing formulations while also facilitating the development of innovative formulations for established pesticides [268,337].

The conditions for producing biopesticides vary for each microbial strain and require meticulous screening to enhance the ultimate performance of the microorganisms in the field. However, there are mainly two physical forms of microbial-based pesticides in the market: dry or liquid [269,339].

#### 11.1.1. Liquid Formulation

Liquid formulations for microbial-based pesticides present diverse options, predominantly featuring microbial cultures or suspensions in water, mineral, or organic oils or polymers [268,340].

One of the most common modes is a suspension concentrate, where solid active ingredients, such as free or immobilized microbial cells, are blended with water or an aqueous solution. These formulations usually incorporate thickeners, dispersants, and wetting agents to maintain stability, prevent sedimentation, and reduce droplet surface tension [341,342].

Emulsions constitute another major category, combining the mixing of immiscible liquids in the presence of an emulsifier. This process results in the dispersion of small droplets within the other phase, with emulsions manifesting as oil in water (O/W) or water in oil (W/O) [343]. Suspo-emulsions combine suspension concentrates with emulsions, offering enhanced efficacy; microorganism concentrations in these formulations typically range from 10% to 40% [76,268,344,345].

These liquid formulations play a crucial role in ensuring stability, efficacy, and ease of application of microbial-based pesticides, offering versatility to suit different needs and application scenarios within biological pest control [268,343].

#### 11.1.2. Solid Formulation

Powder and granular formulations of microbial pesticides offer various options, each tailored to specific production methods and compositions. Microorganisms are intricately blended with carriers and adjuvants in powder formulations, where particles typically range in size from micrometers to granules. This process ensures the uniform dispersion of active ingredients and enhances their stability during storage and application [241,268,337].

Sophisticated techniques like lyophilization and spray drying are usually employed in powder production. Lyophilization dehydrates cells within a protective matrix, safeguarding their viability, while spray drying evaporates water from a liquid matrix, resulting in finely powdered particles [337,346].

Granular formulations, alternatively, often undergo wet granulation processes, where carriers are combined with microbial solutions and carefully dried using air or fluidized bed drying methods [347].

Both wettable powders and water-dispersible granules depend on the integration of dispersants and wetting agents in their formulations to ensure optimal dispersion upon application [348]. Many substances, such as charcoal, sand, talc, kaolin, and alginate, can be applied as carriers for microbial-based biofertilizers and pesticides [349]. These carriers serve as reliable microorganism hosts, effectively controlling soil-borne pathogens while maintaining ecological balance [350,351].

## 12. Pesticides and the Environment

More than 1000 different pesticides are available on the global market [352]. These substances are characterized by their persistence in the environment, and their potential to cause fatalities and chronic diseases, due to their toxicity [81].

According to the FAO, pesticide consumption has increased by 50% globally since the 1990s. However, this has stabilized recently. In 2020, the United States was the largest user of pesticides, with 408 kt applied for agricultural purposes. Following the U.S. are Brazil (377 kt), China (273 kt), Argentina (241 kt), the Russian Federation (91 kt), Canada (79 kt), France (65 kt), Australia (63 kt), India (61 kt), and Italy (57 kt) [353].

From 2018 to 2022, 126 active pesticide compounds were registered for use in sugarcane farms in Brazil, resulting in 947 new products entering the market [354]. There is significant concern about the amount of pesticides used worldwide, as these substances are toxic to life in general, and it is estimated that up to 90% of the applied dose can persist in the environment [81]. Pesticides can persist in soil, be taken up by plants, be carried into freshwater, or be dispersed through the air, depending on various factors such as their chemical nature, quantity used, method of application, soil quality, topography, meteorological conditions, and wind. These factors all influence the ultimate fate of pesticides and their impact on human life (Figure 4).

After being applied to the plant, pesticides can spread in the environment by different routes. It is possible for them to enter the atmosphere by volatilization or absorption by the soil, reaching ground and aquifer water. Pesticides can enter water bodies by runoff from agricultural areas, contaminating the water and harming aquatic habitats. In this context, fish and other aquatic animals are contaminated, disturbing the ecosystem. Human beings can assimilate pesticide-contaminated plants, animals, and water. The breakdown of pesticides by sunlight is known as photo-biodegradation. It is established that the presence of microorganisms in the soil can be used by bioremediation to break down these pollutants. Climate conditions such as wind and soil chemical–physical factors such as pH, humidity, temperature, etc., interfere with pesticide spread and degradation. Possible contamination routes can be seen in Figure 4: 1. plant assimilation; 2. soil absorption/microbial biodegradation, 3. animal contamination (eating contaminated plants); 4. air contamination by volatilization; 5. contamination of the aquatic habitats; 6. aquatic living beings’ contamination; 7. human contamination No. Replace by drawing of Figure 1 and Figure 4.

The Codex Alimentarius, a United Nations organization linked to the World Health Organization (WHO) and FAO, sets standards for the maximum residue limits (MRLs) of pesticides in food through its technical commission: the Codex Committee on Pesticide Residues (CCPR). Over 5000 MRLs have been established for food [355].

### Environmental Residues

Some pesticides are detected in the topsoil, such as glyphosate, pendimethalin, paraquat, chlorpyrifos, and chlorothalonil. In contrast, others are found in deeper soil layers, such as dichloropropene, chlorothalonil, metolachlor, 2,4-D (2,4-Dichlorophenoxyacetic acid), and glyphosate. This characteristic depends on their leachability, solubility, mobility, and volatility [81]. These substances can impact the soil microbiome. For example, a study on 55 orchard soils for 165 pesticides showed 137 pesticide residues and a negative impact on the microbial community, especially Bacillus and Sphingomonas, with an abundance of 43 genera and a reduction of 111 genera [356].

Soil management practices also concern water resources that receive these substances through rain, erosion, and permeation. A study monitoring intensive pasture and sugarcane crop soil management in Brazil over 392 days found that 2,4-D (herbicide) and fipronil (insecticide) were transported to freshwater, reducing the fecundity and female survival of *Ceriophania silvestrii* (aquatic bioindicator) and affecting the larval development of *Chironomus sancticaroli* (sediment bioindicator) and the shoot and root of *Eruca sativa* (phytotoxicity bioindicator) [354]. Furthermore, atrazine is the most detected pesticide on the surface and in drinking water in Brazil, and this pesticide also disturbs the Great Barrier Reef lagoon in Australia [357].

While pesticide exposure in food, soil, and water is relatively well known, air pollution by pesticides requires more attention to define better control standards. Although most pesticide studies focus on soil, attention to air pollution caused by pesticides has increased since 2002. However, these studies are concentrated in Europe and North America, using different methods, which makes the correlations challenging to determine [358,359,360].

Pesticide air diffusion depends on vapor pressure, field dose, meteorological conditions, season, wind, and soil erosion and can be transported to regions far from the source [361]. Volatile compound particle sizes can vary from 2.5 to 10 μm in diameter, and monitoring methods must be efficient to detect them. Wind-eroded soil particles are typically larger than 2.5 μm [358].

The variety and quantity of pesticides widespread in the air are noticeable. For instance, in Germany, between 2014 and 2017, 109 pesticides were observed in the air, 28 of which were forbidden, and glyphosate was present in all samples [361]. Moreover, a study in São Paulo and Piracicaba, Brazil, investigated air particles of pesticides and daily inhalation exposure (DIE), risk assessment, and cancer risk assessment. It found 20 pesticides, 14 of which were quantified. Endosulfan (α and β) and heptachlor, banned since 2013 and 2016, respectively, were detected, indicating recent use. Heptachlor is rapidly converted to heptachlor epoxide, mainly by biodegradation; thus, higher concentrations of heptachlor than heptachlor epoxide suggest recent use [362].

A study on organochlorine pesticides (OCPs) and current-use pesticides (CUPs) in the air of South Africa from 2017 to 2018 revealed 20 OCPs and 15 CUPs in the study areas. A slight increase in OCP concentrations was observed during warmer seasons, but generally, they remained steady, while CUPs showed more fluctuations throughout the year. This research indicated that the concentrations of OCPs, CUPs, and their mixtures did not differ between farms and villages, signaling atmospheric transportation from farms to villages [363].

Research on air pollution by organochlorine pesticides (OCPs) in urban areas of Rio de Janeiro and São Paulo, Brazil, concluded that the population is exposed to high levels of OCPs within the limits of hepatic cancer risk. Detected pesticides included Mirex, Σ-Chlordane, Heptachlor, Endosulfan, HCB, Methoxychlor, Dieldrin, Σ-DDT, and HCH [364].

From 2006 to 2008, air monitoring frequently detected trifluralin, acetochlor, pendimethalin, and chlorothalonil in France. Seasonal influence on pesticide accumulation in the air was noted, with no difference in pesticide profiles between rural and urban areas [365].

Pesticide accumulation in the environment requires constant attention to monitoring, regulation, and remediation technologies to restore or minimize the negative impact on nature.

## 13. Bioremediation of Pesticides by Microorganisms

Bioremediation is an effective technique to reduce or eliminate contaminants by using plants, animals, and microorganisms. Currently, the use of microorganisms for the bioremediation of pesticides is seen as a promising technology due to its cost-effectiveness, reduced toxicity of intermediate compounds compared to physical and chemical remediation methods, and overall environmental benefits [366,367,368].

Studies have shown significant results in the bioremediation of soil and water contaminated with pesticides using microorganisms, which has heightened interest in this technology [367,369]. However, most microbial remediation studies are conducted in laboratory settings, with limited research in complex ecosystems, which reflects natural environmental conditions. Moreover, regulations and guidelines for implementing this technology remain unclear [357,367]. Notably, a successful study demonstrated the effective scale-up of bioremediation from microcosms to mesocosms by combining an *Actinobacteria* quadruple culture, with sugarcane filter cake, to restore lindane-polluted soils [370].

Microbial remediation can be achieved through bioaugmentation using axenic or mixed microbial consortia. Another method is biostimulation, which involves adding biochar, bagasse, surfactants, and plant root exudates to enhance soil microbial activity for pesticide degradation. Biostimulation can improve soil quality and plant development [367]. Natural microbial attenuation is another approach that primarily facilitates pesticide breakdown in soil and water using native microflora [368]. However, environmental contamination often exceeds the natural regeneration capacity because pesticides tend to accumulate in the soil [81].

Innovative approaches, such as biochar-immobilized strains, are being explored for pesticide removal and degradation [371]. For instance, *Serratia marcescens* N80 immobilized on-fungus, substrate-derived biochar effectively degraded thioredoxin-methyl in the soil [372].

Advanced biotechnologies, including metabolomics, metagenomics, transcriptomics, and proteomics, can enhance strategies by identifying degradative pathways and associated genes. Degrading genes, often located on plasmids, transposons, or chromosomes, can be engineered in microbes for improved degradation capabilities [366,373].

There is considerable scope for discovering other microorganisms capable of pesticide bioremediation. Table 8 summarizes well-known microorganisms involved in pesticide bioremediation.

Depending on their enzymatic profiles, microorganisms can utilize pesticides as nitrogen and carbon sources. Generally, microbial consortia are more efficient than single microorganisms in degrading pesticides, as a single species may not possess all the necessary enzymes. Bacterial enzymes involved in pesticide degradation include nitroreductases and esterases. For example, *Pseudomonas* strain ADP uses atrazine as the sole carbon source, with three enzymes facilitating atrazine degradation to CO_2_ and NH_3_. Fungal enzymes, including laccases, oxidoreductases, and peroxidases, are also recognized for their role in pesticide degradation. There is an interest in using purified enzymes from microorganisms rather than whole cells [81].

According to studies, three factors must be considered when selecting the best strategy for microbial pesticide degradation: the type of contaminant, the geological and chemical conditions of the contaminated site, and the kind of microorganism [422].

Pesticides’ chemical and physical properties must be evaluated to determine the most effective remediation strategy or combination of methods to minimize the formation of toxic intermediates. For instance, chlorinating water containing organophosphorus pesticides can form toxic substances that are difficult to remove [423]. Combining microbial bioremediation with other remediation methods may depend on contamination levels, cost-effectiveness, restoration efficiency, and environmental impact [367].

Soil quality, particularly porosity and pH, can affect pesticide adsorption and microbial bioremediation efficiency [81]. Temperature, moisture content, and atmospheric conditions also play significant roles [424]. For example, Yang et al. (2011) found that the optimal temperature for degrading the insecticide carbofuran by *Pichia anomala* HQ-C-01 was 30 °C [380]. Li et al. (2007) found that *Sphingomonas* sp. Dsp-2 had a 58.1% degradation efficiency for chlorpyrifos in acidic soil (pH 4.8) after 7 days, but this increased to 98.7% in alkaline soil (pH 8.7) [425].

Aparicio et al. (2019) constructed an actinomycete quadruple consortium (*Streptomyces* sp. MC1, A5, M7, and *Amycolatopsis tucumanensis* AB0), which effectively removed lindane and Cr (VI) from soil [328]. Xie et al. (2018) conducted a pot experiment combining *Stenotrophomonas* strains and ryegrass (*Lolium perenne*) to remediate DDT and DDE pollution, achieving significantly increased degradation efficiency compared to separate treatments [391].

Eight metabolites were observed from the degradation of metolachlor pesticides by a soil-isolated actinomycete strain, with enhanced degradability in the presence of sucrose, suggesting that the actinomycete did not use metolachlor as its sole carbon source [426].

In a biostimulation study, dissolved organic matter was added to groundwater systems, stimulating the aerobic biodegradation of pesticide 2,4-dichlorophenoxyacetic acid (2,4-D). Complete biodegradation of 2,4-D in groundwater was achieved within 42 days by adding oxygen as an electron acceptor and dissolved organic carbon as a growth substrate [427].

In situ, bioremediation technology was used in this study to remove pesticide contamination from soil around storage facilities. An aerobic bioremediation approach involved direct treatment of contaminated soil using bioaugmentation with three selected active degrading bacterial species: *Stenotrophomonas* sp. (Ps-B strain), *Lysinibacillus fusiformis* (SA-4 strain), and *Enterobacter cloacae* (SB-2 strain). This bioproduct demonstrated a high pesticide decomposition effect, achieving 85.6 to 99.0 ± 0.05% degradation within six months [424].

Immobilized *Pseudomonas putida* on biochar efficiently removed herbicide paraquat from water and supported bacterial degradation into 4,4-bipyridyl and malic acid [332].

Microalgae contribute to pesticide removal through bioadsorption, bioaccumulation, and biodegradation [428]. Microalgae and bacteria can grow together without a symbiotic relationship. *Chlorella vulgaris* collected from the Nile River and immobilized in algae–alginate beads demonstrated 98% and 99% removal of pendimethalin (10 μg/L) and carbofuran (20 μg/L), respectively [378].

Species such as *Chlorella*, *Chlamydomonas*, *Scenedesmus*, and *Nostoc* have documented >90% removal of various pesticides, primarily through biodegradation. Antioxidant enzymes like ascorbate peroxidase, superoxide dismutase, and catalase, as well as the complex structure of microalgae cell walls, play crucial roles in eliminating pesticides and defending against reactive oxygen species [429].

The effects of carbofuran on two microalgae species, *Pseudopediastrum boryanum* and *Desmodesmus communis*, showed that *P. boryanum* completely removed carbofuran from media within 2 days, whereas *D. communis* achieved the same result after 5 days [381].

Surfactants are added to polluted water to enhance the mass transfer of hydrophobic contaminants. Some microbes also produce surfactants to minimize contaminants [422]. As a strategy, adding biosurfactants produced by *Pseudomonas rhodesiae* (C4) increased chlorpyrifos degradation from 39.2% to 51.6% [430].

Microbial remediation offers a promising approach to reducing pesticide contamination. Continued research and development of innovative technologies are essential for improving the efficiency and applicability of bioremediation by microorganisms.

## 14. The Biopesticide Market

According to MarketsandMarkets, the biopesticides market is projected to reach USD 13.9 billion by 2028 from USD 6.7 billion by 2023, at a CAGR of 15.9% during the forecast period in terms of value [431].

The Latin America and North America markets are dominated by microbials overall. The Asian market’s higher use of biochemicals is due to a long history of plant extract and plant growth regulator use. The European Union market has the smallest microbial share due to regulatory policies hindering microbial introduction. Latin America will surpass North America by 2029, driven by Brazil. According to the Brazilian Ministry of Agriculture, Livestock, and Supply, Brazil’s new biological and organic control pesticides were registered in 2020 alone created USD 307 million in revenue [78].

## 15. Conclusions

Bioproducts, particularly biopesticides of microbial origin, demonstrate progress using sustainable and environmentally friendly methodologies. Sourced from bacteria, fungi, baculovirus, and microalgae, these biopesticides provide targeted pest and disease control while significantly reducing environmental impact compared to synthetic pesticides. Various combinations of adjuvants can optimize the action of biopesticides. The study of the best application technique (direct application to the soil, spraying methods, endotherapy, etc.) is fundamental for their effectiveness and focus on the specific target. Microorganisms are potent allies in combating pests in the field in an environmentally friendly way. They can play an essential role in bioremediation decontaminating soil and water. Despite the numerous benefits, MBPs face challenges that must be overcome by a growing body of research pointing to effective applicability with less impact on the environment and humans. Research is underway to overcome these limitations without neglecting the analysis of each scenario and the development of an environment with regulatory laws for the application of biopesticides. This set of factors should lead to broader adoption by a market seeking sustainable agricultural solutions.

## Figures and Tables

**Figure 1 plants-13-02762-f001:**
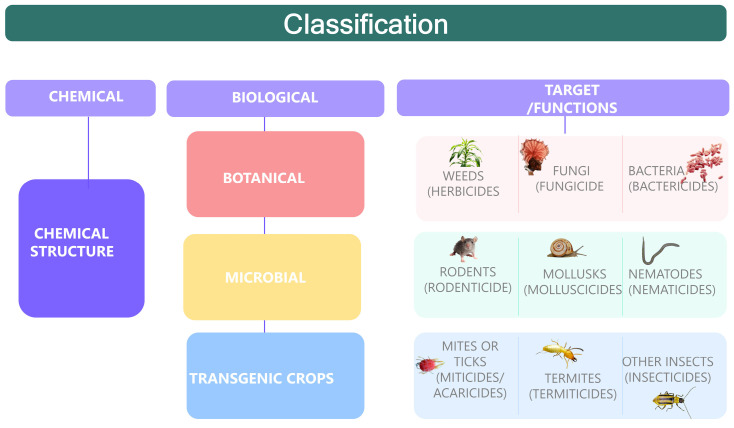
Pesticide classification.

**Figure 2 plants-13-02762-f002:**
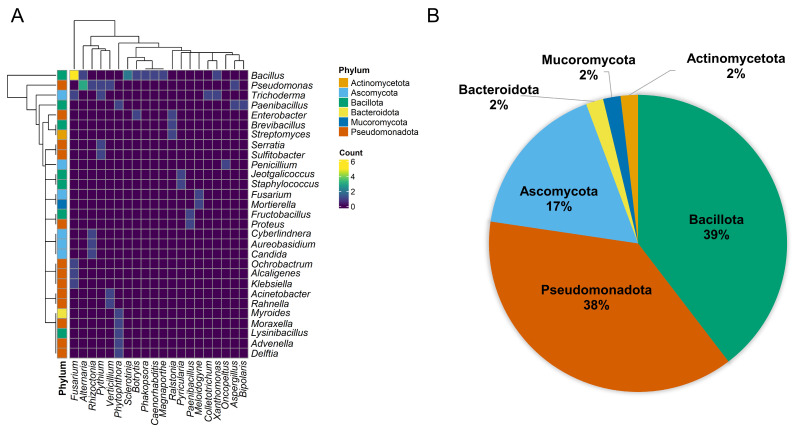
Data obtained from Ref. [126], re-processed in an R environment using the taxizedb [127] and ComplexHeatmap [128] packages. (**A**) Heatmap of occurrences between the genres of antagonists (row) and targets (column); (**B**) pie chart of the sum of the antagonist microorganism phyla.

**Figure 3 plants-13-02762-f003:**
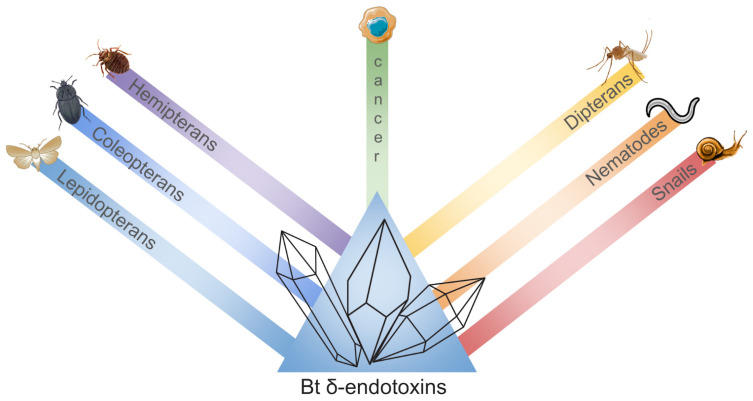
Scheme of the toxicity potential of Bt δ-endotoxins against different organisms.

**Figure 4 plants-13-02762-f004:**
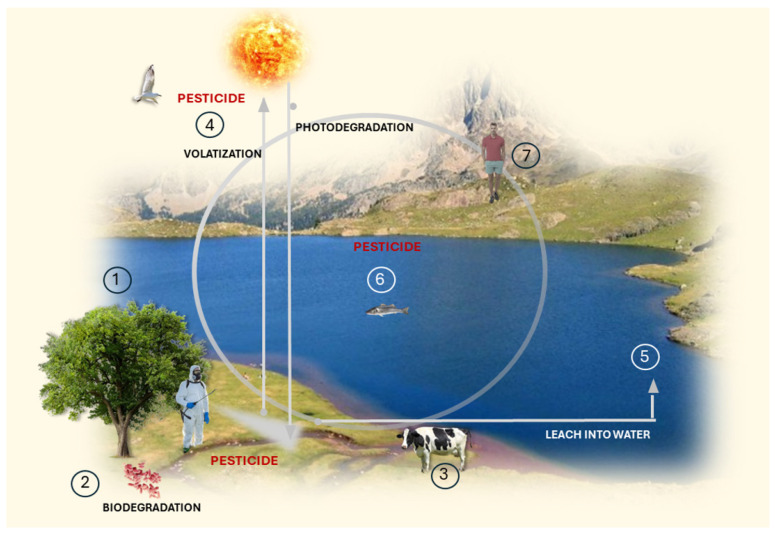
Pesticide cycle in the environment.

**Table 1 plants-13-02762-t001:** Disease, its target, and the respective causative fungi.

Fungi	Disease	Target	References
*Sclerotinia sclerotiorum*	White mold—Sclerotinia rot	Widely distributed soybeans, common beans, cotton, etc.	[23]
*Phytophthora infestans*	Late blight—Phytophthora blight	Potatoes and tomatoes	[24]
*Colletotrichum lindemuthianum*, *C. orbiculare*, *C. capsici*	Anthracnose—Colletotrichum rot	Horticultural, ornamental, and fruit tree crops e.g., alfalfa, almond, apple, blueberry, celery	[25]
*Stagonosporopsis cucurbitacearum* (syn. *Phoma ucurbitacearum*; *Teliomorph*: *Didymella bryoniae*	Gummy stem blight—Didymella rot	Watermelon and other species in Cucurbitaceae.	[26]
*Phomopsis vexans*—seed-borne	Phomopsis—blight	Brinjal (*Solanum melongena*)	[27]
*Alternaria solani*, *Sclerotiorum rolfsii*	Alternaria—rot	Fruit rot in tomatoes, cauliflower	[22,28]
*Pseudoperonospora cubensis*	Mildew and rust	Broad range of monocotyledonous and dicotyledonous plants, economically important crops, and wild plants	[29]
*Macrophomina phaseolina* *Rhizoctonia solani* (soil-borne pathogens)	Charcoal rot or ashy stem blight	Common bean (*Phaseolus vulgaris* L.)	[30]
*Moniliophthora perniciosa*	Witch’s broom	Cocoa	[31]
*Ascochyta rabiei*	Ascochyta—blight	Chickpea (*Cicer arietinum* L.), Lentil (*Lens culinaris*), cowpea (*Vigna unguiculata*), pea (*Pisum sativum*), and the common bean (*Phaseolus vulgaris*	[32]
*Fusarium* sp.	Crown rate Stem-end rot Mango Papaya Wheat and barley	Banana Avogado, potato Leaf spot Fruit rot *Fusarium* head blight, FHB	[33]

**Table 2 plants-13-02762-t002:** Same pesticide with chemical classification [71].

Chemical Classification	Examples of Pesticides
Organochlorines	Diclorodifeniltricloroetano (DDT), dichlorodiphenyldichloroethylene (DDE), dichlorodiphenyldichloroethane (DDD); hexachlorocyclohexane (HCH), such as lindane; cyclodienes, including aldrin, dieldrin, endrin, heptachlor, chlordane, and endosulfan; toxaphene; mirex and chlordecone
Organophosphates	Malathion, parathion, diazinon, chlorpyrifos, dichlorvos, phorate
Carbamates	Thiobencarb, propoxur, molinate, disulfiram (Antabuse), pyridostigmine, methiocarb, carbaryl
Pyrethroids	Permethrin, deltamethrin, cypermethrin, fenvalerate, bifenthrin

**Table 3 plants-13-02762-t003:** Active compounds from plants used as biopesticides.

Plant	Structure Used	Active Compound	Target	Refs.
*Acorus calamus* L.	Leaf, rhizome, and stem	A-asarone and β-asarone,	Fungi: *Microsporum gypseum*, *Penicillium marneffei*, *Trichophyton rubrum* Insect: *Sitophilus zeamais*	[102,103]
*Curcuma longa* L.	Root stem	Curcumin	Caterpillar: *Spodoptera frugiperda*, *Spodoptera litura*	[104,105]
*Ocimum basilicum* L.	Leaf	Major: Linalool and Cinnamic acid methyl ester	Fungi: *Alternalia solani*, *Alternaria heveae*, *Phytophthora infestans*	[106,107]
*Azadirachta indica*	Mainly in seeds	Azadirachtin, meliantriol, salannin, desacetyl salannin, nimbin, desacetyl nimbin, and nimbidin	Insect: Dictyoptera, Orthoptera, Heteroptera, Isoptera, Lepidoptera, Diptera, Coleoptera, Homoptera, Siphonaptera, and Hemiptera	[108,109]
*Artemisia* *herba-alba* Assso		Cineole, thujone, camphor, davanone	Insect: *Callosobruchus maculatus* Fungi: *Fusarium moniliforme*, *F. oxysporum*, *F. solani*	[110,111]
*Mentha piperita*	Leaves	Menthol, menthone, iso menthone, neomenthol, acetyl menthol, pulegone, methyl acetate	Fungi: *Botrytis cinerea* Insect mites, mosquito larvae	[112,113]
Garlic (*Allium sativum*)	Bulbs	Allicin	Fungi: *Drechslera tritici-repetis*, *Bipolaris sorokiniana*, and *Septoria tricidi*	[114]

**Table 4 plants-13-02762-t004:** A list of some commercial products, crops, nuclear virus polyhedroviruses (NPVs) or granuloviruses (GVs), and target insect-available baculovirus formulations.

Commercial Product and Country	Crop	Virus	Target Insect	Reference
GemStar LC (Mexico, Brazil, EUA USA, Argentina) Virin-HS (Russia) DOA BIO V2 (Brazil)	Cotton, pepper soybean, tomato, and other rowed crops	NPV	*Heliothis* sp. *Helicoverpa armigera* (cotton bollworm, corn earworm, tobacco budworm, corn earworm)	[155,156,157]
Madmex^TM^ (Switzerland) Cyd-X (EUA) USA Virosoft CP4 (EUA USA, Canada, European countries) Carpovirusine^TM^ ( France) granupon^TM^ (Switzerland) Granusal^TM^ (Germany) Virin-CyAP (Russia) Carposin (EUA USA, China, European Union) Carpovirus SC (Argentina)	Pear, apple, and walnut	NPV	*Cydia pomenella* (codling moth)	[158,159,160]
Baculo-soja (Brazil), Multigen (Brazil), BaculovirusNitral (Brazil), Coopervirus SC (Brazil), Protege (Brazil)	Soybeans	NPV	*Anticarcia gemmatalis* (velvet been caterpillar)	[161]
Gusano Biological (Brazil) VPN-80^TM^ (Guatemala)	Alfalfa vegetable	NPV	*Autograph acalifornica* (beet armyworm alfalfa looper and several other lepidopteran species)	[162,163]
Spodopterin (Brazil) DOA BIO V3 (Brazil)	Cotton, soybeans, tomato, peanut, cabbage, corn, and tobacco	NPV	*Spodoptera litura*	[164]
Capex 2 (Brazil)	Apple and pear	GV	*Adoxophyes orana*	[160,165,166,167]
Agrovir (Brazil)	Maize	GV	*Agrotis segetum*	[160,168]
Disparvirus (Brazil) Gypchek (EUA) Virin-ENSH (Brazil)	Forestry	NPV	*Lymantria dispar*	[169]
GemStar (Mexico) Biotrol (Brazil) Elcar (Brazil)	Cotton, vegetables	NPV	*Helicoverpa zea*	[170]
VPN-82 (Guatemala) VPN-Ultra (Guatemala)	Horticulture	NPV	*Spodoptera albula*	[171]
Spod-X (Brazil) Vir-ex (Brazil) Spod-XLC (Mexico DOA BIO V1(Brazil) Ness-A (Argentina) Ness-E (Argentina ARG) Otienem-S^TM^ (Brazil)	Vegetables	NPV	*Spodoptera exigua* (beet armyworm)	[148,161]
Spodopterin^®^ (several countries, with emphasis on Brazil) Littovir^®^ (Brazil)	Cotton, corn	NPV	*Spodoptera littoralis*	[166,172]
DOA BIO V3 (Brazil)	Vegetables	NPV	*Spodoptera littoralis* (cotton leafworm)	[166]
Mamestrin (Brazil)	Oilseed rapes	NPV	*Mamestra brassicae*	[162,166]
Leconti-virus (Brazil)	Forestry	NPV	*Neodiprion lecontei*	[173]
Neochek-S (South Korea)	Forestry	NPV	*Neodiprion sertifer*	[174]
Biocontrol I Virtuss (Brazil)	Forestry	NPV	*Orgyia pseudotsugata*	[169]
Virin-ABB (Brazil)	Forestry	NPV	*Hyphantri acunea*	[148]

**Table 5 plants-13-02762-t005:** Advantages and disadvantages of fungi-based pesticide.

Advantages	Disadvantages
Environmentally friendly	Slow action
Safe for workers	Susceptibility to abiotic stress factors: temperature and exposure to UV radiation affect the survival of fungi
Wide host range	Limited shelf life
No chemical residues in food	Difficult in standardization of rest phase of conidia
Sustainable production	Higher costs *

* Costs of production.

**Table 6 plants-13-02762-t006:** Secondary metabolites produced by some fungal endophytes.

Metabolite (s)	Producer Fungi	Fungal Pathogen(s)	Refs.
Bicolorin D	*Saccharicola bicolor*	*Sclerotinia sclerotiorum*	[191]
Brefeldin A	*Cladosporium* sp.	*Aspergillus niger*	[192]
Pretrichodermamide A	*Trichoderma harzianum* *Epiccocum nigrum*	*Ustilago maydis*	[193]
Methyl dichloroasterrate	*Aspergillus capensis*	*Botrytis cinerea*, *Monilinia fructicola*, *Sclerotinia sclerotiorum*, *Sclerotinia trifoliorum*	[194]
Trichodermin	*Trichoderma brevicompactum*	*Rhizoctonia solani* *Fusarium solani*	[195]
Versicolorin	*Aspergillus versicolor*	*Colletotrichum musae*	[196]

* Costs of production.

**Table 7 plants-13-02762-t007:** Spectrum of application of entomopathogenic fungi-based biopesticides against plant pathogens and insect pests.

Entomopathogenic Fungi	Plant Pathogens	Insect Pests	Ref.
*Beauveria bassiana*	*Rhizoctonia solani* *Gaeumannomyces graminis* var. *tritici* *Botrytis cinerea*, *Alternaria alternata*	*Macrosiphum euphorbiae* (Hemiptera) *Myzus persicae* (Hemiptera) *Hyposidra talaca* (Lepidoptera) *Helopeltis theivora* (Heteroptera)	[207,208,209,210]
*Metarhizium* anisopliae	*Fusarium graminearum* *Botrytis Cinerea*	*Helopeltis theivora* (Heteroptera) *Phyllotreta striolata* (Coleoptera) *Monochamus alternatus* (Coleoptera)	[211,212,213,214,215]
*Paecilomyes farinosus* *Isaria javanica*	*Colletotrichum gloeosporioides* *Phytophthora capsici*	*Galleria mellonella* (Lepidoptera) *Planococcus ficus* (Hemiptera) *Pyrrhocoris apterus* (Hemiptera)	[198,216,217,218]
*Lecanicillium lecanii*, (formerly known as *Verticillium lecanii*)	*Cactodera estônica* *Hemileia vastatrix*	*Empoasca flavescens* (Hemiptera) *Diaphorina citri* (Hemiptera) *Plutella xylostella* (Lepidoptera)	[219,220]

**Table 8 plants-13-02762-t008:** Microorganisms associated with pesticide bioremediation.

Compounds	Microorganisms	Environment	References
Amide	*Paracoccus huijuniae* sp. nov.	Culture medium	[374]
Atrazine	*Acinetobacter lwoffii* DNS32 by biochar-immobilization	Soil	[375]
	*Arthrobacter* sp. ZXY-2	Water	[376]
	*Streptomyces* sp. atz2, *Aspergillus niger* AN 400, *Pseudomonas* sp., *Achromobacter* sp., *Basidiomycetes*, *Fusarium* sp., *Microcystis novacekii*	Culture medium	[357]
	*Stereum hirsutum*	Soil	[377]
Atrazine, desethylatrazine, and desisopropylatrazine	*Pleurotus ostreatus* INCQS 40310	Culture medium	[357]
Atrazine, molinate, simazine, isoproturon, propanil, carbofuran, dimethoate, pendimethalin, metolachlor, and pyriproxyfen	*Chlorella vulgaris*	Soil and water	[378]
Carbofuran	*Burkholderia cepacia* PCL3	Soil	[379]
	*Pichia anomala* HQ-C-01		[380]
	*Pseudopediastrum boryanum* and *Desmodesmus communis*	Water	[381]
Chlorophenoxyacetic acids	*Dichomitus squalens*	Culture medium	[382]
Chlorpyrifos	*Bacillus megaterium* HM01 by biochar-immobilization	Soil and water	[383]
	*Cupriavidus nantongensis* X1T	Culture medium	[384]
	*Flavobacterium* EMBS0145	Soil	[385]
	*Rhodotorula glutinis*, *Rhodotorula rubra*	Culture medium and tomato fruit	[386]
Cypermethrin	Consortium by *Bacillus zhanjiangensis* TJTB48, *Bacillus pseudofirmus* TJTB58, and *Oceanobacillus kimchii* TJTB66 by biochar-immobilization	Soil	[387]
DDT	*Chryseobacterium* sp. PYR2		[388]
	*Serratia marcescens* NCIM 2919		[389]
DDT, DDE	*Sphingobacterium* sp. D6		[390]
	*Stenotrophomonas* sp. DXZ9		[391]
Diazinon	*Candida pseudolambica*	Culture medium	[392]
	*Scenedesmus obliquus*, *Chlamydomonas mexicana*, *Chlorella vulgaris*, and *Chlamydomonas pitschmannii*	Water	[393]
Dibutyl phthalate	*Bacillus siamensis* T7 by biochar-immobilization	Soil	[394]
Dicofol	*Microbacterium* sp. D-2		[395]
Difenoconazole	Consortium by *Aspergillus flavus* and *Bacillus cereus* S1	Culture medium	[396]
Dimethachlon	*Providencia stuartii* JD by biochar-immobilization	Soil	[397]
Dimethomorph	*Bacillus cereus* WL08 by biochar-immobilization	Soil and water	[398]
Diuron	*Micrococcus* sp. PS-1	Soil	[399]
	*Pluteus cubensis* SXS 320	Culture medium	[357]
Diuron and bentazon	*Trametes versicolor*	Water	[400]
Endosulfan	*Cladosporium oxysporum*	Culture medium	[401]
β-endosulfan	*Pandoraea* sp.		[402]
Hexachlorocyclohexane (HCH)	*Sphingobium ummariense*		[403]
	*Sphingomonas* sp. NM05	Soil	[404]
	*Penicillium griseofulvum*	Culture medium	[405]
Imidacloprid	*Mycobacterium* sp. MK6		[406]
Lidane	*Thermobifida cellulosilytica* TB100	Soil	[407]
	*Azotobacter chroococcum* JL102		[408]
	*Streptomyces* sp.	Culture medium	[357]
Metribuzin	Consortium by *Bacillus tequilensis* AQ2, *Bacillus aryabhattai* AQ3, *Bacillus safensis* AQ4, and *Rhodococcus rhodochrous* AQ1 by biochar-immobilization	Soil	[409]
Organophosphate	*Flavobacterium* sp. ATCC 27551	Culture medium	[410]
Oxyfluorfen, 2,4-D, diuron, and sulfentrazone	*Bradyrhizobium* sp. BR 3901		[411]
Paraquat	*Pseudomonas putida*	Water	[412]
	*Hypholoma dispersum* ECS-705	Culture medium	[413]
Polyacrylamide	*Klebsiella* sp. PCX	Soil	[414]
Pyrethroid	*Raoultella ornithinolytica* ZK4	Culture medium	[415]
Pyridine	Consortium by *Enterobacter cloacae complex* sp. BD17 and *Enterobacter* sp. BD19	Water	[416]
Quinclorac	*Cellulosimicrobium cellulans* D	Soil	[417]
Sulfonylurea	*Trichoderma harzianum*	Culture medium	[418]
Tebuconazole	*Alcaligenes faecalis* WZ-2	Soil	[419]
Thiamethoxam	*Phanerochaete chrysosporium*		[420]
Thifensulfuron-methyl	*Serratia marcescens* N80		[372]
Triclocarban	*Pseudomonas fluorescens* MC46	Water	[421]

## Data Availability

Not applicable.

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
