# Peer review of "Agricultural Pest Management: The Role of Microorganisms in Biopesticides and Soil Bioremediation"

_plants, 2024, doi:10.3390/plants13192762_

Round 1

Reviewer 1 Report

Comments and Suggestions for Authors

Overall the paper is unorganized and not focused. The title is misleading. The paper contains a laundry list of topics and history lessons related to pesticides broadly and their classification, formulations, and biological control. Many of the important terms, including biological control, are not defined. Everything is surface level and does not offer anything novel to the field. The actual coverage of microbial biopesticides is incomplete and there needs to be more on the availability, application and efficacy of currently available products if the review intends to focus on this method for pest management. There are many tertiary, and irrelevant sections that need to be removed to focus the paper and more details related to the topic at hand provided. The best part was the writings around bioremediation of pesticides by microbes, but again, this is adjacent to the topic of microbial biopesticides and rather an application of microbes to combat residues from conventional pesticides—NOT what the title claims to cover.

Comments on the Quality of English Language

There are some grammatical errors that need to be addressed, but the paper is not ready for that yet.

Author Response

 Reviewer 1

Overall the paper is unorganized and not focused. The title is misleading. The paper contains a laundry list of topics and history lessons related to pesticides broadly and their classification, formulations, and biological control. Many of the important terms, including biological control, are not defined. Everything is surface level and does not offer anything novel to the field. The actual coverage of microbial biopesticides is incomplete and there needs to be more on the availability, application and efficacy of currently available products if the review intends to focus on this method for pest management. There are many tertiary, and irrelevant sections that need to be removed to focus the paper and more details related to the topic at hand provided. The best part was the writings around bioremediation of pesticides by microbes, but again, this is adjacent to the topic of microbial biopesticides and rather an application of microbes to combat residues from conventional pesticides—NOT what the title claims to cover.

 The objective was to write a comprehensive and updated review. The reviews in the literature about this topic report similar items, such as classification, which must be included in the current manuscript for the review to be complete. As suggested by the reviewer, the title was modified, and structural modifications were made to the manuscript text to increase the focus and information. No section was removed because of other reviewers' comments about it.

Reviewer 2 Report

Comments and Suggestions for Authors

A review of Microbial Biopesticides for Sustainable Plant Agricultural Pest Management is relevant. The paper appears well planned, relatively well written, and represents a good contribution to science literature. Although the manuscript is well written, I strongly recommend carefully revising grammatical and syntactical errors because there are many grammatical errors throughout the paper. In general, the manuscript needs to take care of many details, the order of some concepts is important, they are noted in the manuscript.

Author Response

Reviewer  2

A review of Microbial Biopesticides for Sustainable Plant Agricultural Pest Management is relevant. The paper appears well planned, relatively well written and represents a good contribution to science literature. Although the manuscript is well written, I strongly recommend carefully revising grammatical and syntactical errors because there are many grammatical errors throughout the paper. In general, the manuscript needs to take care of many details, the order of some concepts is important. They are noted in the manuscript.

Thank you for the corrections and comments. Repeated concepts and the order of subitens were corrected,   and the suggestions have been incorporated into the text.

  1. Entomopathogenic fungi and nematodes are not so specific, so I suggest including a discussion about them.

We included two topics about  Entomopathogenic fungi (subsection 6.5) and  6.6. Entomopathogenic nematodes and their bacterial symbionts  subsection  6.6 ) in 6.0 producing microorganisms   

  1. Table 4

 The countries were included.

  1. Table 5

 The table was  modified, and  more information about this was added to the manuscript

  1. 3. Remotion of the Subsections Pesticide and the Environment and Bioremediation of Pesticides by Microorganisms

 All corrections have been implemented based on the reviewer's feedback. Subsection 11, Pesticides and the Environment cannot be removed as other reviewers have deemed it essential and requested modifications in this section.

Soil contamination by pesticides is a problem for farmers in pesticide-contaminated areas. Bioremediation is necessary for pesticide-contaminated soil before applying a biocontrol agent. Another possibility is applying a bioremediation agent together with microorganisms used as biopesticides.

Reviewer 3 Report

Comments and Suggestions for Authors

Dear authors,

The manuscript titled “Microbial Biopesticides for Sustainable Plant Agricultural Pest Management” is interesting including a lot of information”. However, after careful consideration, I would like to suggest you some additions.

-In all sections you should add more information and examples about the use of bioherbicides.

-It would be useful to add a section related to the challenges, disadvantages and limitations of biopesticides

-It would be useful to report some comparisons in terms of effectiveness between conventional pesticides and biopesticides

-Examples of circular economy and production of biopesticides could be cited, too.

Comments on the Quality of English Language

Minor revisions are required.

Author Response

Reviewer 3

Dear authors,

The manuscript titled “Microbial Biopesticides for Sustainable Plant Agricultural Pest Management” is interesting including a lot of information”. However, after careful consideration, I would like to suggest you some additions.

  1. In all sections, you should add more information and examples about the use of bioherbicides.

 Thank you, and we added more information concerning the bioherbicide use

  1. It would be useful to add a section related to the challenges, disadvantages, and limitations of biopesticides

 New sections were made with more information on this topic.

  1. It would be useful to report some comparisons in terms of effectiveness between conventional pesticides and biopesticides

  More information was added in the review.

  1. Examples of circular economy and the production of biopesticides

 We added a new section

Round 2

Reviewer 2 Report

Comments and Suggestions for Authors

The manuscript was corrected as requested

Academic Editor Notes

The manuscript has been substantially improved with most of the suggestions made by the reviewers and editors. Only the following minor fixes are needed:

1. P13. L 501: VIZ correct for VIP

The correction was made.

  1. To ensure consistency throughout the text, please use the abbreviation uniformly as either "VIP" or "Vip”

 Okay, the abbreviation was uniformized.

  1. Table 4: correct Argentina.

The correction was made.

  1. Table 4: correct spelling of the species: seprate genus

from species for all the species in the table

The error was corrected.

  1. Table 4: In italics Lymantria dispar

The error was corrected.

  1. Table 4: “Pear, Apple, and walnut”: “walnut” in capital letter

Okay the correction was made.

  1. Table 4. The clouding Moth: Correct for: codling moth

 Olay.

  1. Table 4: Correct for velvet been caterpillar

 The correction was made.

  1. P19 L646: Herbicide: not in capital letter

The correction was made.

  1. P20 L652 EFP is not defined

The correction was made.

 EFPis the abbreviation of  Entomopathogenic fungi

  1. P20 L657 “Verticillium lecanii are significative” correct italics in common text.

The correction was made.

  1. P27. L 946. Correct numeration in subtitle: 8.1. Surfactants and Carriers  

 All numbers, titles, and subtitles of the review were corrected.
